# Do Language Models Use Their Depth Efficiently?

**Róbert Csordás**   **Christopher D. Manning**   **Christopher Potts**
Stanford University, Stanford, CA, USA
rcsordas@cs.stanford.edu   {manning,cgpotts}@stanford.edu

## Abstract

Modern LLMs are increasingly deep, and depth correlates with performance, albeit with diminishing returns. However, do these models use their depth *efficiently*? Do they compose more features to create higher-order computations that are impossible in shallow models, or do they merely spread the same kinds of computation out over more layers? To address these questions, we analyze the residual stream of the Llama 3.1, Qwen 3, and OLMo 2 family of models. We find: First, comparing the output of the sublayers to the residual stream reveals that layers in the second half contribute much less than those in the first half, with a clear phase transition between the two halves. Second, skipping layers in the second half has a much smaller effect on future computations and output predictions. Third, for multihop tasks, we are unable to find evidence that models are using increased depth to compose subresults in examples involving many hops. Fourth, we seek to directly address whether deeper models are using their additional layers to perform new kinds of computation. To do this, we train linear maps from the residual stream of a shallow model to a deeper one. We find that layers with the same relative depth map best to each other, suggesting that the larger model simply spreads the same computations out over its many layers. All this evidence suggests that deeper models are not using their depth to learn new kinds of computation, but only using the greater depth to perform more fine-grained adjustments to the residual. This may help explain why increasing scale leads to diminishing returns for stacked Transformer architectures.

## 1 Introduction

Large Language Models (LLMs [1, 2, 3, 4, 5]) have improved rapidly in recent years, and one significant correlate of these improvements is their increasing *depth* as measured by number of Transformer layers (Fig. 1). This scaling relationship would seem to follow from the structure of these LLMs: they predominantly use a stacked Transformer structure [6], which lacks recurrence across layers, and thus the number of computation steps they can perform is constrained by their depth. In theory, greater depth should enable them to perform more complex computations by building on top of the representations computed in previous layers. Deeper models should have the capacity to be more compositional, leading to better reasoning, math capabilities, and generalization.

However, it is unclear whether these models are using their depth efficiently. On the one hand, Petty et al. [7] find that increasing depth does not help with compositional generalization, Lad et al. [8] show that, apart from the first and last layers, models are robust to layer skipping and swapping neighboring layers (see also Sun et al. [9]), and Gromov et al. [10] were able to remove half of the layers from the network without significantly affecting performance on MMLU (but not for math). On the other hand, interpretability research often finds evidence for complex mechanisms spanning multiple layers [11, 12], suggesting that models can represent more complex operations as their depth increases. This paper is an exploration of this tension. Our primary question: do deeper LLMs use their depth to compose more features to create higher-order computations that are impossible in shallow models, or do they merely spread the same kinds of computation out over more layers?

39th Conference on Neural Information Processing Systems (NeurIPS 2025).

We focus on the Llama 3 series of models and supplement these findings with secondary analyses of the Qwen 3 series. Following the findings of previous work, we mostly focus on the math domain, which shows the greatest sensitivity to perturbations. Our analysis consists of five parts:

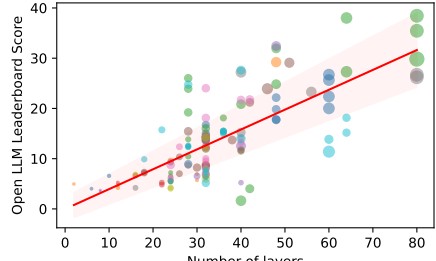

1. We analyze the norm of the residual stream and compare it to each individual sublayer's output, as a measure of that layer's contribution. We find a drop in this quantity in the second half of the model.

2. We measure the effect that skipping a layer has on all successive layers' computations. These analyses show that, for the *current* output token, nearly all layers seem to be important, but the computations in the second half of the layers depend minimally on each other. In contrast, for *future* token predictions, skipping the second half of the layers has a minimal effect. This suggests that these layers are mainly refining the final probability distribution rather than computing reusable sub-results. We verify this using Logitlens [14], which shows a drop in KL-divergence

Figure 1: Performance of 132 Open LLM Leaderboard [13] base models as a function of depth. Colors represent different model families; dot size is proportional to parameter count. Linear regression in red, with 95% confidence interval. Depth is a significant predictor even in regressions that control for other scale-relevant factors (App. B). Deeper models generally perform better.

and a sharp increase in the top prediction overlap with the final layer, starting around the same layer as the importance for future predictions decreases.

3. We analyze multihop questions and difficult math questions, and look for evidence of deeper computations for more complex examples. However, our analysis shows the contrary: the layer's sensitivity to previous layers seems to be independent of example complexity. Analyzing individual examples with both causal interventions and integrated gradients [15] shows that the input tokens remain important until the middle of the network, and later tokens in the multi-step computation are not delayed to later layers, suggesting that no composition is happening.

4. We train linear maps from each layer of a shallower model to each layer of an independently trained deeper one, sharing the same vocabulary. By measuring the prediction error of the linear maps, we can measure the correspondence of the layers of the two models. This shows a diagonal pattern, indicating that the deeper model merely spreads out the computation through more layers instead of doing more computation in the later layers.

Overall, these findings suggest that current LLMs underutilize the second half of their layers. Rather than using their depth to learn more complex computations, they instead simply spread out the same kind of computation through an increasing number of layers, taking smaller computation steps and devoting the second half of the network to iteratively refining the probability distribution of the current token. We conclude with an exploration suggesting that MoEUT [16] might use its layers more efficiently.

## 2 Background

All the models we analyze are pre-layernorm Transformers [6, 17]. A Transformer layer $l$ is constructed as follows:

$$\boldsymbol{a}_l = \text{SelfAttention}_l(\text{Norm}(\boldsymbol{h}_l)) \tag{1}$$

$$\hat{\boldsymbol{h}}_l = \boldsymbol{h}_l + \boldsymbol{a}_l \tag{2}$$

$$\boldsymbol{m}_l = \text{MLP}_l(\text{Norm}(\hat{\boldsymbol{h}}_l)) \tag{3}$$

$$\boldsymbol{h}_{l+1} = \hat{\boldsymbol{h}}_l + \boldsymbol{m}_l \tag{4}$$

Here, $\boldsymbol{h}_l \in \mathbb{R}^{T \times d_{\text{model}}}$ is the residual stream and $\boldsymbol{a}_l, \boldsymbol{m}_l \in \mathbb{R}^{T \times d_{\text{model}}}$ are the outputs of the SelfAttention and MLP layers, respectively, where $T$ is the length of the current input sequence and $d_{\text{model}}$ is the width of the residual stream. Norm$(\cdot)$ is some token-wise normalization, traditionally layer normalization [18], but usually replaced with RMSNorm [19]. We call SelfAttention$_l(\cdot)$ and MLP$_l(\cdot)$ *sublayers*.

The residual stream is initialized with $\boldsymbol{h}_0 = \text{Embedding}(\boldsymbol{x})$, where $\boldsymbol{x} \in \mathbb{N}^T$ is the sequence of input token indices. The output probability distribution, or *prediction*, is $\boldsymbol{y} = \text{softmax}(\hat{\boldsymbol{y}})$, where $\hat{\boldsymbol{y}} =$

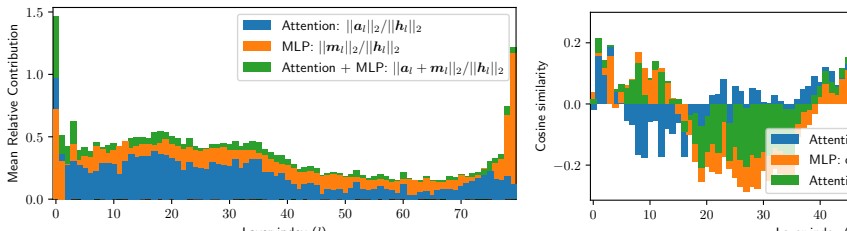

| (a) Relative norm of (sub)layer contributions | (b) Cossims of (sub)layer contributions and the residual |

Figure 2: Influence of layers and sublayers on the residual stream for Llama 3.1 70B. (a) Norm of contributions relative to the residual stream. A sharp drop is visible near the middle; later layers change the residual much less, with the exception of the last few layers. (b) Cosine similarity between the contributions and the corresponding residual shows a phase change at the middle of the network.

$\text{Norm}(\boldsymbol{h}_L)\boldsymbol{W}^{out}$ are the output logits, $L$ is the number of layers in the network, $\boldsymbol{W}^{out} \in \mathbb{R}^{d_{\text{model}} \times V}$ are the weights of the output classifier, and $V$ is the size of the vocabulary.

Note that in pre-layernorm Transformers, the interaction of each sublayer with the residual is additive, as shown in Eq. 2 and 4. Thus, we can quantify the *contribution* of layer $l$ to the residual stream as $\boldsymbol{a}_l + \boldsymbol{m}_l = \boldsymbol{h}_{l+1} - \boldsymbol{h}_l$. The contribution of the sublayers is $\boldsymbol{a}_l$ for the attention, and $\boldsymbol{m}_l$ for the MLP.

## 3 Experiments

Most of the experiments presented in the main paper are performed with Llama 3.1 70B [20], using NDIF and NNsight [21]. Unless noted otherwise, the results are computed on 10 random examples from GSM8K [22]. In bar plots, each bar starts from 0 (no stacking). The main results are also shown in the appendix on different models, including Llama, Qwen [23], and OLMo 2 [24]. In Sec. 3.1, we measure how the layers and sublayers contribute to the residual stream. In Sec. 3.2, we use causal interventions to measure the effect of layers on downstream computations. In Sec. 3.3 we show that deeper or otherwise more complex computations do not influence the number of layers that have a causal effect on the prediction model. In Sec. 3.4 we train linear projections to find the correspondence between the layers of an independently trained shallow and deep Qwen model. Our exploration in Sec. 3.5 suggests that MoEUT [16] might use its layers more efficiently, especially when not modeling the question.[1]

### 3.1 How do the Layers Interact With the Residual Stream?

Since all interaction with the residual stream in pre-layernorm Transformers is additive, it is expected that the norm of the residual, $||\boldsymbol{h}_l||_2$, will grow in later layers. At initialization, the norm of the output of each sublayer ($||\boldsymbol{a}_l||_2$ and $||\boldsymbol{m}_l||_2$) is identical in expectation due to the normalization layer at their input. Thus, later layers contribute less than the earlier ones: it is harder for them to change the direction of the residual. During training, the model can learn to compensate for this growth by increasing the norm of the weights in later layers. However, most models are trained with weight decay, which explicitly discourages such growth. Residual growth was previously observed in the context of outlier features [25] and Universal Transformers [16]. Here, we seek to use this technique to gain an initial high-level understanding of how much each layer contributes.

**The relative contribution of sublayers.** We measure the $L^2$ norm of the residual $||\boldsymbol{h}_l||_2$ and attention and MLP contributions ($||\boldsymbol{a}_l||_2$ and $||\boldsymbol{m}_l||_2$) in all layers of Llama 3.1 70B [20]. We observe the expected rapid growth of the residual (Fig. 15, in Appendix). However, the growth of the sublayer outputs seems to be slower. To zoom in on this, in Fig. 2a, we measure the mean relative contribution of each (sub)layer ($\frac{||\boldsymbol{a}_l+\boldsymbol{m}_l||_2}{||\boldsymbol{h}_l||_2}$, $\frac{||\boldsymbol{a}_l||_2}{||\boldsymbol{h}_l||_2}$ and $\frac{||\boldsymbol{m}_l||_2}{||\boldsymbol{h}_l+\boldsymbol{a}_l||_2}$). This shows a consistent contribution in the first half of the network, with a significant drop around the middle. The drop is especially pronounced in the attention layers. The only exception is the last few layers, where the contributions seem to grow again.

---

[1]Our code is public: https://github.com/robertcsordas/llm_effective_depth

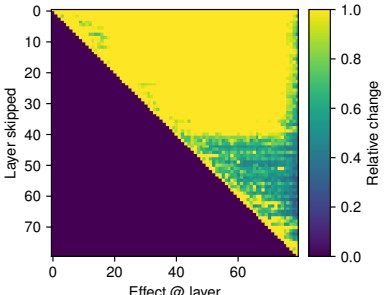
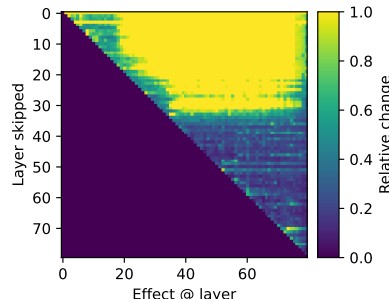

(a) Effect of skipping a layer on later layers' contributions in the *all* timesteps.

(b) Effect of skipping a layer on later layers' contributions in *future* timesteps.

Figure 3: The maximum relative change in the layer's contribution when a previous layer is skipped, Llama 3.1 70B on GSM8K [22]. (a) Shows the maximum effect on the future computations for all tokens in the sequence, including the *current* token, while (b) isolates the effect only for the maximum of the *future* tokens. The range is limited between 0 and 1. (a) The second half of the layers has a weaker effect on future computations compared to the first. Because of the low influence on future layers in (a), but high importance for prediction (Fig. 17a in the Appendix), the second half of the layers seems to perform mostly independent, but important, computations to refine the current predicted probability distribution. This is supported by the findings of Fig. 4. (b), which shows that the second half has little effect on the future tokens, indicating that they are not computing reusable subresults.

**Measuring the cosine similarity between the residual and sublayer contributions.** To dig deeper into each layer's contribution to the overall computation, we measure the average cosine similarity between the output of different layers and sublayers and the residual. This is defined by cossim($\boldsymbol{m}_l + \boldsymbol{a}_l, \boldsymbol{h}_l$) for the layer, and cossim($\boldsymbol{a}_l, \boldsymbol{h}_l$) and cossim($\boldsymbol{m}_l, \boldsymbol{h}_l + \boldsymbol{a}_l$) for the SelfAttention and MLP components, respectively, where cossim($\boldsymbol{x}, \boldsymbol{y}) = \frac{\boldsymbol{x} \cdot \boldsymbol{y}}{||\boldsymbol{x}||_2 ||\boldsymbol{y}||_2}$. Where features are orthogonal to each other, zero cosine similarity corresponds to writing a new feature to the residual, negative values correspond to erasing features, and positive values mean strengthening an existing feature.[2]

We show the results in Fig. 2b. The first layer has near-zero cosine similarity, suggesting that these layers are primarily integrating context from neighboring tokens. This is followed by a mostly positive phase where features are being refined. The rest of the first half of the layers largely tends to erase the residual. Around the middle of the network, a sharp phase transition is visible: the model starts strengthening existing features instead of erasing information. Interestingly, this position corresponds to the drop in the layer's contributions observed in Fig. 2a.

The changes in the relative contributions and the cosine similarities are high-level indicators of a possible phase change. In the following, we investigate this more closely.

## 3.2 How do the Layers Influence Downstream Computations?

**Causal intervention for measuring the layers' importance for the downstream computation.** In Section 3.1, we analyzed the general characteristics of the residual stream. Here, we turn to pairwise interactions between layers using interventions. Which layer's computation is influenced by a previous layer? In order to address this question, we use the following procedure. First, we run a prompt through the model and log the residual $\boldsymbol{h}_l$. Second, we run the same prompt again, but this time we skip layer $s$, by setting $\bar{\boldsymbol{h}}_{s+1} := \bar{\boldsymbol{h}}_s$, and we log the residual of the intervened model $\bar{\boldsymbol{h}}_l$. Third, we measure the relative change in the contribution of layer $l > s$: $\frac{||(\boldsymbol{h}_{l+1} - \boldsymbol{h}_l) - (\bar{\boldsymbol{h}}_{l+1} - \bar{\boldsymbol{h}}_l)||_2}{||\boldsymbol{h}_{l+1} - \boldsymbol{h}_l||_2}$. We take the maximum of this metric over the sequence and multiple prompts. We choose maximum because some of the later layers might only be used rarely, when a deep computation requires it. We

---

[2]Two limitations of this method: (1) Transformers are hypothesized to make heavy use of features in superposition [26, 27, 28, 29], so this intuition might not fully transfer to practice; (2) if the model combines erasing/strengthening and writing new features, the new features will not show up in the cosine similarity metric.

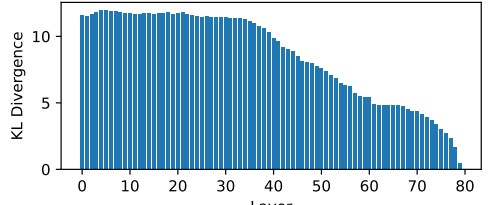
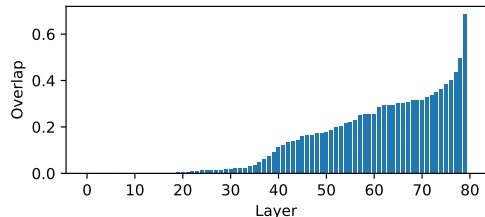

(a) KL divergence between the model's prediction and Logitlens applied to each layer.

(b) Overlap in top-5 tokens from the model's prediction and Logitlens applied to each layer.

Figure 4: Comparing Logitlens on different layers to the final prediction. (a) KL-divergence. (b) Overlap in the top-5 predicted tokens. Both show that later layers are devoted primarily to refining the output probability distributions, rather than to performing new kind of computation.

also compare the model output probabilities: $||\boldsymbol{y} - \bar{\boldsymbol{y}}||_2$. We chose this non-standard metric because it provides clearer visualizations compared to KL divergence.

We show the results in Fig. 3a. We can see that, unlike the early layers, the layers in the second half of the model have a low influence on the computations performed in the later layers. However, Appendix Fig. 17a shows that these late layers are equally important for the output predictions, indicating that they perform important, but independent, computations.

In summary, these layer skipping experiments indicate that the layers in the first half of the network are integrating information and potentially building on each other's output, while the second half refines the output probability distribution based on the information already present in the residual.

**Measuring layer importance for *future* predictions.** To investigate this effect more deeply, we perform a variant of the previous experiment to measure the effect on the *future* tokens when skipping layers for earlier tokens. We do this by sampling a position $1 < t_s < T - 1$, skipping the layer only for token positions $t \leq t_s$, and measuring the effect only on positions $t > t_s$. The results are dramatic: as Fig. 3b shows, the second half of the network barely has any effect on future computations, except for some special layers at the very top of the network. Furthermore, their effect on future predictions is also significantly less than for the layers in the first half of the network (Appendix Fig. 17b).

**What happens in the second half of the network?** To validate our hypothesis that the second half of the layers refines the probability distribution of the current prediction, we apply Logitlens [14] to the residual and measure the KL divergence between its prediction and the final prediction of the model. The results are shown in Fig. 4a. Furthermore, we measure the overlap between the set of top-5 predictions from Logitlens and the final distribution in Fig. 4b. Both show the same picture: the prediction refinement seems to start at the same position at the same phase transition as when the layers do not influence the future predictions anymore, where the cosine similarities change sign, and the layer's importance decreases. All of these observations support our hypothesis: the second half of the network is merely doing incremental updates to the residual to refine the predicted distribution.

**Localizing Circuits.** A similar method can also be used to discover layers that build on the contributions of previous layers directly. In order to do so, we can measure the change in future layers' contributions when removing the target layer's contribution from their input. In contrast to the previous experiments, we do not propagate this change to later layers. Specifically, to measure the effect of layer $s$, we set $\bar{\boldsymbol{h}}_l := \boldsymbol{h}_l - \boldsymbol{h}_s$ for all $l \geq s$, and measure the relative change in the layer's contribution: $\frac{||(\boldsymbol{a}_l+\boldsymbol{m}_l)-(\bar{\boldsymbol{a}}_l+\bar{\boldsymbol{m}}_l)||_2}{||(\boldsymbol{a}_l+\boldsymbol{m}_l)||_2}$. Fig 5 shows the results. Bright spots indicate layers that build on each other's features. When isolating the effect on future tokens only, it is possible to localize multi-layer, multi-token mechanisms similar to induction heads (Fig. 5b).

### 3.3 Do Deeper Problems Use Deeper Computation?

If a network is doing compositional computation, it has to break the problem down into sub-problems, solve the sub-problems, and combine their solutions. Because of the lack of recurrence, Transformers only see the results of computations in successive layers. We would therefore expect to see that

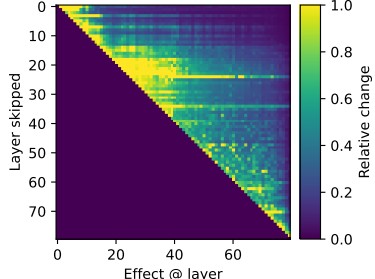 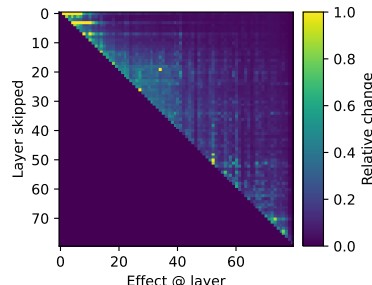

(a) Local effect of layer on later layers' contributions in the *all* timesteps.

(b) Local effect of layer on later layers' contributions in *future* timesteps.

Figure 5: Analyzing the direct local effects between pairs of layers of Llama 3.1 70B on GSM8k [22]. The heatmaps highlight layer pairs with direct effects on each other. Unlike Fig. 3, the effects are not propagated to future layers. For each layer $s$, the plot shows future layers that build on the representation computed by $s$. (a) Effects on all tokens, highlighting all possible circuits. (b) Effect on future tokens. The sparse, bright spots indicate multi-layer, multi-token mechanisms, such as induction heads. Note that interacting layers are not necessarily spatially close to each other.

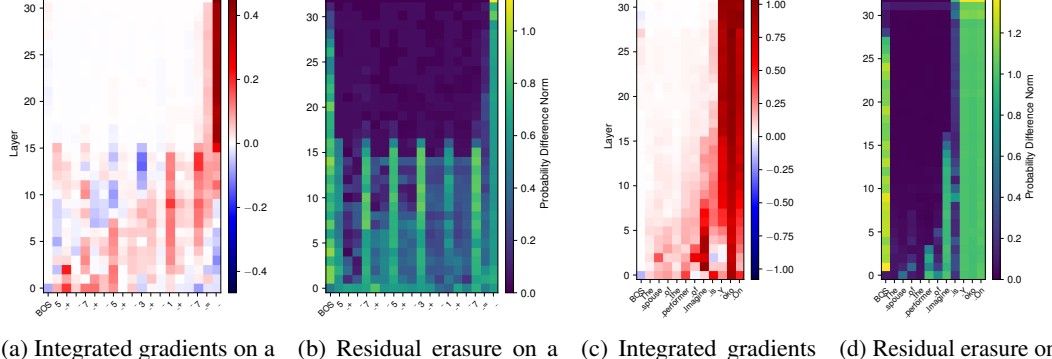

(a) Integrated gradients on a math question.

(b) Residual erasure on a math question.

(c) Integrated gradients on a two-hop reasoning.

(d) Residual erasure on a two-hop reasoning.

Figure 6: The effect of individual computation steps on Llama 3.1 8B. (a,b) Basic math question. (c,d) Two-hop reasoning. Note that the answer is 4 tokens long in this case, providing a stronger gradient signal. (a,c) Integrated gradients. (b,d) The probability distribution change ($||\boldsymbol{y} - \bar{\boldsymbol{y}}||_2$) when erasing the residual of a given token in a given layer. This shows until when the information of a token is used. In both cases, the second half of the model shows minimal effect. Moreover, in arithmetic, later hops of computation do not use more depth, indicating that no composition is happening.

problems with deeper computation graphs use more layers in the Transformer. Additionally, later steps of a composite computation should be executed in later layers, so that they can receive the results of earlier subproblems as inputs. Are models in fact organizing their computations this way?

**Residual erasure interventions.** We check if the models are using more depth for later computation based on individual prompts. We compute two metrics: one is Integrated Gradients [15], where we compute the gradient on all answer tokens, but not on the prompt. The second metric is the maximum prediction norm change ($||\boldsymbol{y} - \bar{\boldsymbol{y}}||_2$) among the answer tokens when the residual is changed to be uninformative. We call this intervention "residual erasure". It shows until which layer the information from a given token is used. This erasure intervention is done for each possible position $t$ and layer $l$, by setting $\bar{\boldsymbol{h}}_{l+1}[t] := \tilde{\boldsymbol{h}}_l$ while keeping the rest of the tokens unchanged ($\bar{\boldsymbol{h}}_{l+1}[t'] := \boldsymbol{h}_{l+1}[t']$ for all $t' \neq t$), and the visualizing the effect. The uninformative residual, $\tilde{\boldsymbol{h}}_l$, is the average of the residual in a given layer, computed on multiple examples (in our case on GSM8K) over batch and time. $\boldsymbol{h}_l$ is the residual from the original, non-intervened model on the same prompt, and $[\cdot]$ is the indexing operation for accessing a single element of a vector or matrix.

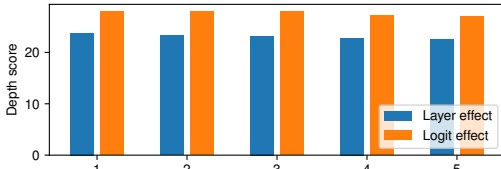
(a) Depth score on different difficulty levels of MATH

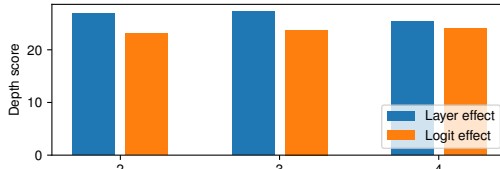
(b) Depth score in function of hops on MQuAKE

Figure 7: Depth score: the weighted average of layer index with its importance. Importance is measured based on the effect on later layer contributions on future predictions (see Fig. 3b for more details). (a) MATH dataset [30]. The x-axis is the difficulty level defined by the dataset. (b) MQuAKE [31]. The x-axis is the number of hops in the question. The depth of computation the model performs is independent of the problem difficulty and the number of hops in the input problem.

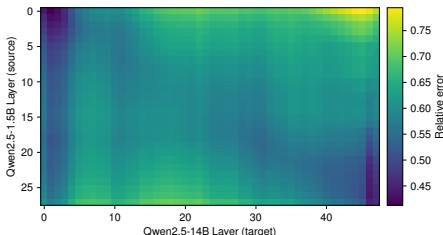

Figure 8: Linear map accuracies predicting Qwen 2.5 14B activations from Qwen 2.5 1.5B. A clear diagonal trend is visible: layers with the same relative position map to each other the best, indicating that deeper models "spread out" the same kind of computation, computing the prediction in smaller steps.

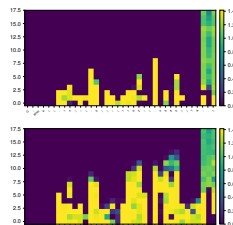

Figure 9: Comparing the residual erasure in Transformers (top) and MoEUT (bottom) on an example from DeepMind Math [32], when trained without modeling the question. MoEUT uses more of its depth, and its depth seems to be more input-dependent. See Fig 42 and Fig 43 in the Appendix for more, zoomed-in examples.

Fig. 6 summarizes our results. For arithmetic, in both metrics, we see that all tokens are equally important until the middle of the model. For two-hop reasoning, the picture is somewhat less clear, but the second half of the model still shows no sign of computation that is useful for the predictions.

**The Depth Score.** In order to more systematically verify if the models are using deeper computations of later operations, we measure the max influence of layers on later layers and the output on future positions, similarly to Fig. 3b and 17b. For each layer, we take the mean effect on all future layers in future tokens, thus reducing the maximum effect matrix to a vector with a dimension equal to the number of layers. For the separation point between the past and future tokens, we sample multiple positions from the answer. Additionally, we further reduce these "effect size" vectors by computing $d = \sum_{l=1}^{L} l \frac{e_l}{\sum_m e_m}$. Here, $e_l$ is the importance of layer $l$ computed by any of our previously defined metrics. We call $d$ the "depth score". This score increases if the model uses the later layers more.

We analyze two datasets: MQuAKE [31], which consists of multi-hop questions with a known number of hops, and the MATH dataset [30], which consists of complex math problems with different difficulty levels. We consider these difficulty levels as a proxy for the required computation depth. We measure the above metrics on 20 different random examples for each expected depth. If more hops or more complex questions use more depth, we expect to see that the contribution of the later layers increases with complexity, and that should be reflected in our metrics. Fig. 7 summarizes these analyses. We see no evidence of deeper computations with increased difficulty. (More detailed plots can be found in Fig. 38 in the Appendix.)

### 3.4 Do Deeper Models Do Novel Computation?

Are deeper models performing computation that is not present in the shallower ones because of a lack of layers, or is it "stretching out" the same kind of computation over more layers, performing smaller

steps at a time? The first kind of effect would be preferable, because the deeper model is capable of combining more features and composing more subresults in theory. If such novel computation is present, it should be hard to predict from the activations of the shallow model, and the number of predictable layers should be close to the number of layers in the shallow model. In order to verify this, we take two pretrained models with different layer counts ($L_1$ and $L_2$, with $L_1 < L_2$), and train $L_1 L_2$ different linear probes to map every point in the residual of the shallower model $\boldsymbol{h}_l^1$ to each representation in the deeper model $\boldsymbol{h}_m^2$. This requires that the two models use the same tokenizer, and, for reliable results, they should be trained independently, instead of being distilled from each other. Because of GPU memory limitations, we decided to use the Qwen 2.5 [33] series of models because of their more modest size compared to the Llama models. Concretely, we train a linear map for each pair of layers of Qwen 2.5 1.5B and 14B. We measure the relative prediction error $\frac{||\boldsymbol{h}_l^{14\mathrm{B}} - f_{lm}(\boldsymbol{h}_m^{1.5\mathrm{B}})||_2}{||\boldsymbol{h}_l^{14\mathrm{B}}||_2}$, where $f_{lm}(\cdot)$ is the linear map from layer $m$ of the small model to $l$ of the big model. We do this for each $m, l$ layer pair and plot it in Fig. 8. Although some ranges of layers seem to be easier to predict than others, a clear diagonal pattern is visible. This shows that the big model is more likely a "stretched out" version of the shallow model, rather than one that does entirely new computations.

### 3.5  Is the Pretraining Objective or the Model Responsible for using Fixed Depth?

To explore what causes the models to use fixed depth for each computation step regardless of the problem, we trained standard Transformers and MoEUT [16] on the DeepMind Math dataset [32]. We test MoEUT because Universal Transformers [34] enable easy "transfer" of knowledge from early layers to later ones, thanks to parameter sharing. Additionally, LLMs are trained on free-form text and should model everything regardless of whether they are a "question" (unpredictable) or an "answer" (often predictable given the correct circuits). We also wanted to test whether modeling this uncertainty plays an important role, so we trained both models with and without learning to predict the question.

We use the 244M parameter baseline and MoEUT models from the paper [16], without modifications. We perform the residual erasure intervention on four examples (see Appendix Sec. D.5), and display the most important results in Fig. 9 (with details in Figs. 42 and 43 in the Appendix). We can clearly see that the models that were trained to *not* model the uncertain question use more of their layers in processing the answer, probably because they do not have to spend their capacity on modeling the high-entropy probability distribution of the unpredictable questions. Surprisingly, MoEUT successfully achieves this even when learning to model the question, although the effect is more pronounced without it. This confirms the advantage of the shared-layer models. Interestingly, while all models have good interpolation performance, their extrapolation capability differs drastically. For example, MoEUT on the "Mul/Div Multiple Longer" split has an accuracy of 36% if modeling the question is enabled, and 63% if it is not. The difference for standard Transformers is less dramatic (41 vs. 48%).

Interestingly, all models trained from scratch show increased depth with deeper computation steps. The effect is more pronounced with the MoEUT models, especially if the question is not learned. This is in contrast to what we found for most examples when fine-tuning a Llama model (App. D.5), confirming that fine-tuning might not be enough to change the pretrained model's behavior fundamentally.

## 4  Related Work

Lad et al. [8] discuss the four stages of inference: detokenization, feature engineering, prediction ensembling, and residual sharpening. The authors show that, in early and late layers, the model is sensitive to layer skipping, but not in the middle. In one of their main claims, the authors show that throughout the layers, the attention to the previous five tokens gradually decreases. However, this might mean that the attention integrates further away context, or might attend to a broader set of tokens. The authors also show a slightly reduced contribution of attention compared to the MLP in later layers, but the reduction is gradual and not dramatic. In contrast, we use interventions to directly show that later layers have minimal effect on future predictions. The authors also do not study the effects of input complexity on processing depth, nor the effect of increased model depth. Skean et al. [35] discover an information bottleneck in the middle of autoregressive Transformers, and show that the intermediate representations often outperform the final ones for downstream tasks.

Multiple prior works have examined the effect of layer interventions, such as skipping, swapping, or parallelization [9, 10, 8]. In general, they find that models are remarkably robust to such interventions

on most of the tasks. The notable exception found by all papers are math-related tasks, such as GSM8K. This corresponds to the intuitive expectation that math should require composing subresults. This requires a large number of layers in Transformers, proportional to the depth of the computation graph. This is the reason why we decided to focus on math-related tasks, but we found no evidence for such deeper compositions.

For additional related work, please refer to Appendix F.

## 5   Discussion

**Is the second half of the layers wasteful?** Our experiments show that a significant proportion of layers are not used to construct higher-level features for downstream computations, but only refine the final probability distribution. Although matching the real probability distribution very closely might be useful for language modeling, for solving downstream tasks, and also in practical models after instruction tuning, only the top few token probabilities are important. It is therefore surprising, and seems wasteful, that models spend half of their capacity distribution-matching instead of further integrating information and doing more composition. The independence of operations performed by later layers also implies that all the information should already be present in the residual simultaneously. Thus, the residual width ($d_{\text{model}}$) might be an important bottleneck.

**The consequence of fixed depth computations.** Using causal interventions, we show that more complex problems do not cause the computation to shift to deeper layers. Although LLMs lack explicit adaptive computation time mechanisms [36, 37], they can, in theory, learn to control the amount of computation implicitly. The complete lack of any evidence for dynamic computation is surprising. This means that the models do not break down the problem into subproblems, solve them, and recompose them to solve the full problem, but instead process everything with a fixed circuit on a fixed computation budget. It is unclear how such fixed-depth solutions can generalize to the vast compositional structure of the world, without learning different circuits for each situation. The long-tailed distribution of such mechanisms might help explain the diminishing returns of increased scaling.

**The connection to Chain of Thought.** Chain of Thought [38] avoids the lack of compositional processing in the rich representations of the residual stream by outsourcing it to the input/output space. At inference time, this results in full recurrence with discretization between steps. Other advantages of this approach include supervision on the internal steps (either from pretaining or during fine-tuning), and the discretization denoising intermediate computation steps. On the down side, the model cannot learn to adaptively think more whenever it is needed but not reflected in the training data (e.g., arithmetic operations are rarely written out in papers). The state is also limited to discrete symbols.

**Consequences for Latent Thinking approaches.** Recently, a method for "thinking" in the latent space [39] was proposed, relying on recurrent processing in the residual stream to avoid some of the limitations of Chain of Thought. If the insensitivity of computation depth to input complexity is the consequence of the pretaining objective, such methods are fundamentally flawed. On the other hand, if the reason is the architecture, these approaches might provide the solution. Thus, determining the reason and finding a possible solution to this problem is an important research direction.

## 6   Conclusion

In this paper, we quantify the amount of processing done by each layer and the interaction between layers in pretrained language models. Using causal interventions, we found that in the second half of their layers, these models do not build further on intermediate representations computed in earlier layers. This casts doubts on the efficiency of the mechanisms learned by these models, raising concerns about the importance of later layers. We also found that the depth of processing does not change as a function of input complexity. This indicates that the models do not dynamically build on the output of previous computations to perform more complex ones. This casts doubts on recent approaches that aim to get models to "think" in their latent space. We also show that, when learning a linear map between two models with different layer counts, the layers at the same relative positions correspond to each other the most, indicating that the deeper model merely spreads out the same type of computation that the shallower one uses. Our exploratory look at the recently proposed MoEUT model indicates that it might use its layers more efficiently than Transformers. Our findings call

for research on better architectures and training objectives that can leverage the deep layers more efficiently.

## 7 Acknowledgements and Funding Transparency Statement

The authors would like to thank David Bau, Shikhar Murty, Houjun Liu, Piotr Piękos, Julie Kallini, and the members of the Stanford NLP Group for helpful feedback and discussions at various stages of this project. We are thankful to Adam Belfki, Emma Bortz, and the NDIF engineers for their support with experimental infrastructure. This research is supported in part by grants from Google and Open Philanthropy. Christopher D. Manning is a CIFAR Fellow.

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

# Appendix

## A    Limitations

The paper is a case study on multiple Llama and Qwen models. Although the findings seem to be robust for these models, they might not hold for other model types.

Although the paper shows that models do not do computation that depends on the complexity of the problem, it does not answer the question of how models solves the problems without such dependence. This is an important direction for future work.

Sec. 3.4 relies on a single model pair, because of the expense of training $L_1 L_2$ big linear classifiers while also keeping the models running. The findings should be verified with more resources on different and deeper models.

Sec. 3.5 relies on manual study of individual trained models. An automatic metric that measures the correspondence of the computation to the parse tree should be developed. However, this is a nontrivial task that we leave for future work.

Nevertheless, we believe that our paper provides novel evidence for the high level inner workings of LLMs. We hope that this inspires a future direction of research on how to improve them.

## B    Model Depth as a Factor Shaping Performance

Fig. 1 briefly analyzes the role of model depth in shaping model performance, using a dataset of 132 base models on the Open LLM Leaderboard [13]. To more deeply explore this relationship, we fit a linear regression predicting performance using scale-relevant factors of these models: depth, model dimensionality, and feed-forward dimensionality. (Total parameters is also a potential predictor, but it is highly correlated with these other variables.) Depth is a highly significant variable in this model ($p < 0.0001$). This result is highly robust to rescaling of the independent variables and including model family as a hierarchical grouping factor. Thus, it seems clear that making models deeper does make them better, even though the models themselves do not seem to use their depth efficiently.

## C    Robustness Analysis

To justify our choice of a relatively low number of samples (10-20, depending on the experiment) and a single dataset (GSM8k), we present results for more samples and for a different dataset (Math). We show the influence of layers and sublayers on the residual stream, equivalent to Fig. 2, in Fig. 10. Furthermore, we show the layer skipping effects, equivalent to Fig. 3, in Fig. 11, and the Logitlens-based output similarity, corresponding to Fig. 4, in Fig. 12. These analyses show that our conclusions are robust both with respect to the dataset choice and the number of examples.

To show the variability between individual examples, we show 4 examples contributing to Fig. 3b in Fig. 13. Their variability justifies our choice to take the maximum over them to quantify the overall importance of the layers.

## D    Results on Other Models

The performance on the HELM Lite benchmark of a few important models is shown in Fig 14. Performance improves with the number of layers, similar to the Open LLM leaderboard (Fig. 1).

### D.1    How do the Layers Interact With the Residual Stream?

We show the absolute and relative contributions of the sublayers to the residual stream for Llama 3.1 8b in Fig. 18, for the Qwen 3 series of models in Fig 19 and for OLMo 2 models in Fig 20. We show cosine similarities of the sublayer's contributions and the residual stream for all Llama and Qwen models that we tested in Fig. 21 and for the OLMo models in Fig. 22. The results are similar to our findings in Sec. 3.1.

We show the cosine similarity of the neighboring layers ($\mathrm{cossim}(\boldsymbol{h}_l, \boldsymbol{h}_{l+1})$) in Fig. 16. All neighboring layers have very high cosine similarity, often close to 1, which is a consequence of the known

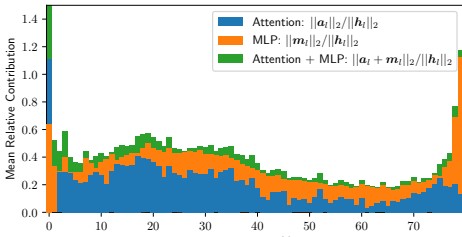

(a) Math dataset: Relative norm of (sub)layer contributions

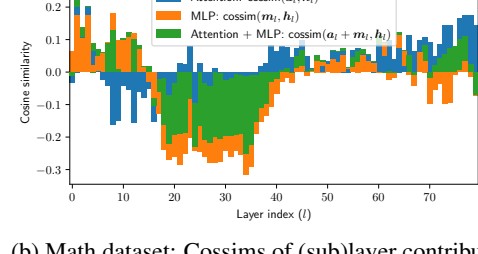

(b) Math dataset: Cossims of (sub)layer contributions and the residual

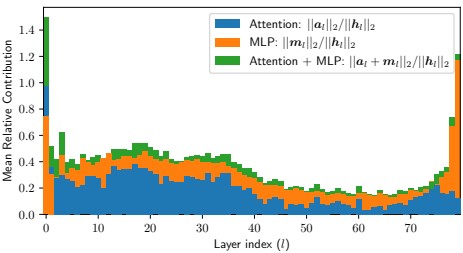

(c) 50 examples: Relative norm of (sub)layer contributions

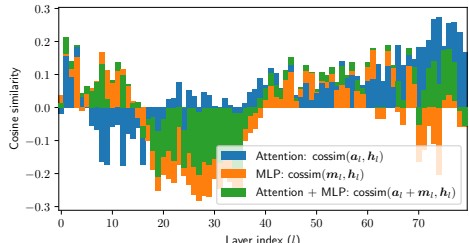

(d) 50 examples: Cossims of (sub)layer contributions and the residual

Figure 10: Robustness analysis: influence of layers and sublayers on the residual stream for Llama 3.1 70B. (a,c) Norm of contributions relative to the residual stream. (b,d) Cosine similarity between the contributions. Math dataset (a,b) and 50 examples on GSM8K (c,d) are shown. The findings are consistent with Fig 2.

anisotropy of Transformers [40]. However, as Fig. 2b and Fig. 21 show, comparing the cosine similarity of the residual stream and the contributions of the layers and sublayers reveals a rich structure.

## D.2 How do the Layers Influence Downstream Computations?

Here, we show the effect of individual layers on later layers in future timesteps and on future token outputs for multiple models. Fig 23 shows the effect on Llama 3.1 8B and 70B, while Fig. 24 shows the Qwen 3 series of models, Fig. 25 shows the instruction-tuned Llama 3.1 70B, and Fig. 26 shows the OLMo 2 models. It can be seen that instruction tuning has no influence on the model's behavior. In the Qwen 3 series of models, the effect seems to be less pronounced, but still present. OLMo 2 behaves very similarly to Qwen 3. The findings agree with our discussion in Sec. 3.2.

We show the local layer interactions for other Llama models in Fig. 27, for the Qwen 3 models in Fig. 28 and for OLMo 2 in Fig. 29.

Additional Logitlens results are shown for the other models in Fig. 30 and Fig.31.

Qwen 3 32B (Fig. 24e) seems to display an additional interesting effect: early layers seem to work independently, not building on each other's representation from the previous timesteps. The integration across time seems to start at around layer 40. Interestingly, once this point in the network is reached, all previous computations seem to be important. Fig 28f shows the existence of a single layer that integrates most of the information from the past. This layer composes features of many previous layers.

We show the cosine similarities between the contributions and the residual for all tested models in Fig. 21.

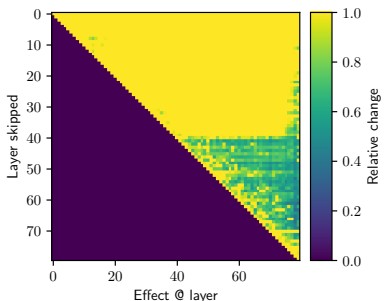

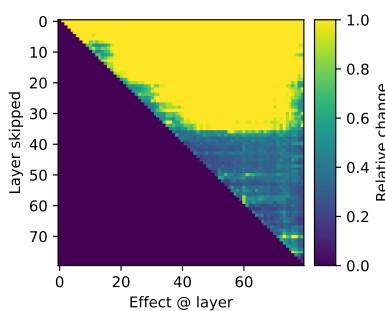

(a) Math Dataset: Effect of skipping a layer on later layers' contributions in the *all* timesteps

(b) Math Dataset: Effect of skipping a layer on later layers' contributions in *future* timesteps.

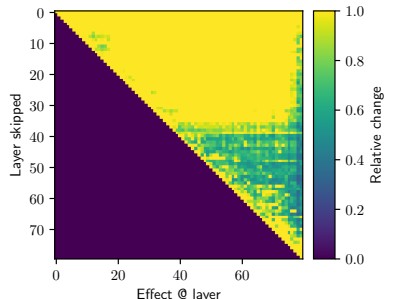

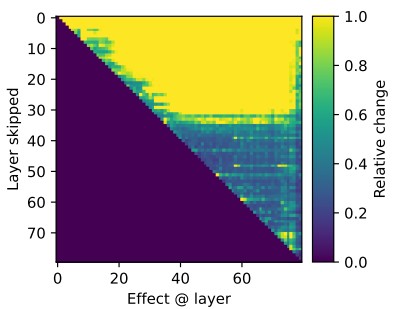

(c) 50 examples: Effect of skipping a layer on later layers' contributions in the *all* timesteps

(d) 50 examples: Effect of skipping a layer on later layers' contributions in *future* timesteps.

Figure 11: Robustness analysis: effects of layer skipping on all tokens (a,c) and future tokens (b,d), on the Math Dataset (a,b) and with 50 examples on GSM8K (c,d). All variants agree with our findings in Fig. 3.

### D.3 Do Deeper Problems Use Deeper Computation?

Additional residual erasure experiments are shown for Llama 3.1 70B in Fig. 32, for the Qwen 3 series of models in Fig. 33 and for OLMo 2 in Fig. 34. Findings for the 70B Llama models are identical to those discussed in Sec. 3.3. Qwen models use more layers, but they also seem to use a fixed number of layers independently of the computation depth, indicating that they are not building on subresults from previous computation steps. OLMo seems to have weaker depth dependence than the other models. All findings are consistent with Sec 3.3.

We show the depth score on MATH and MQuAKE datasets for Llama 3.1 8B in Fig. 35 and for the Qwen 3 series of models in Fig. 33. The findings are identical to what we discussed in Sec. 3.3.

### D.4 Do Deeper Models Do Novel Computation?

In Sec. 3.4 we used Qwen 2.5 1.5B and 14B models instead of the newer Qwen 3 series. The reason for this is twofold: first, given that we had to train $L_1 L_2$ different linear maps of substantial size, we chose small models to be able to fit both models simultaneously on a single A6000 GPU. Second, given this size limitation, the difference in the layer count of the Qwen 2.5 is higher than the viable options from the Qwen 3 series.

### D.5 Does Finetuning Cause the Model to Use Deeper Computations?

We finetune all parameters of Llama 3.2 3B on the arithmetic splits of the DeepMind Math dataset [32], with batch size 64, for 10k steps, with a warmup of 100 steps followed by a constant learning rate of $2 * 10^{-5}$. At the end of the training, we perform integrated gradients and residual erasure experiments on both the base model, which was the starting point of the finetuning process, and the

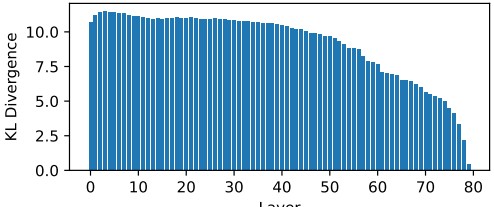
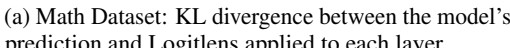

(a) Math Dataset: KL divergence between the model's prediction and Logitlens applied to each layer.

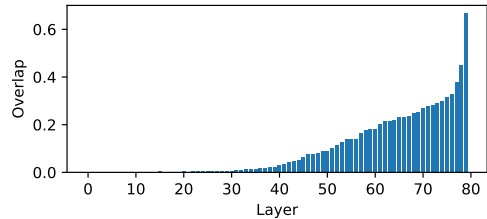

(b) Math Dataset: Overlap in top-5 tokens from the model's prediction and Logitlens applied to each layer.

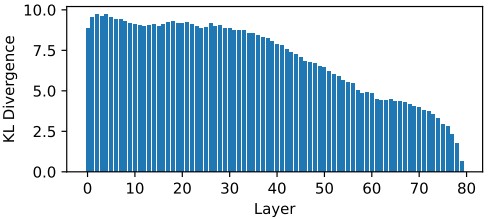

(c) 50 examples: KL divergence between the model's prediction and Logitlens applied to each layer.

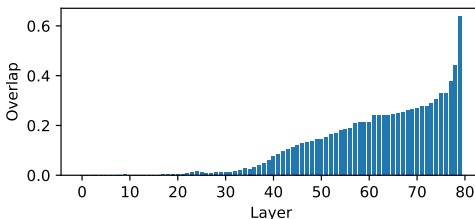

(d) 50 examples: Overlap in top-5 tokens from the model's prediction and Logitlens applied to each layer.

Figure 12: Robustness analysis: comparing Logitlens on different layers to the final prediction. (a,c) KL-divergence. (b,d) Overlap in the top-5 predicted tokens. Math dataset (a,b) and 50 examples on GSM8k (c,d) are shown. The findings are consistent with Fig. 4

final model. We also include the instruction-tuned version of the model as a control. We measure the maximum effect on later layers and predictions of future tokens and find that fine-tuning seemingly helps to increase the computation depth significantly (Fig. 39). However, by looking deeper at individual instances, we reveal that the effect is mostly marginal: only the last 1–2 tokens are affected before the prediction, and the residual erasure experiment shows no significant difference in the point when they become unimportant, indicating that they are used in parallel (Fig. 40). We also tried applying the loss only to the answers, but in contrast to pretraining (Sect. 3.5), it seems to have no effect (Fig. 41).

# E    Details on the DeepMind Math Training

We fine-tune/pretrain our models on the arithmetic subset of the DeepMind Math dataset. These are all the files in the train set that begin with the string "arithmetic_". To feed an example to the network, use the template "Q: question A: answer". To fill the context window of the model, we concatenate multiple such examples with a whitespace between. We never break examples if they do not fully fit the context window; the end of the window is padded as needed.

# F    Extended Related Work

Previous studies on the residual stream suggested that ResNets behave like an ensemble of shallow networks [41]. Gurnee and Tegmark [42] showed that in Transformer language models, linear probe accuracy increases rapidly in the first half of the model, and the improvements become marginal in the second half. A phase transition was also previously observed around half of the model, where the activations transition from sparse activations to dense [43, 44]. Logitlens [14] was also observed to provide meaningful predictions from around the middle of the network. The growth of the residual stream was previously observed in the context of outlier features [25], where they were hypothesized to be one of the causes of the outlier features, and for large-scale Universal Transformers [16], where they present an obstacle to mechanism reuse.

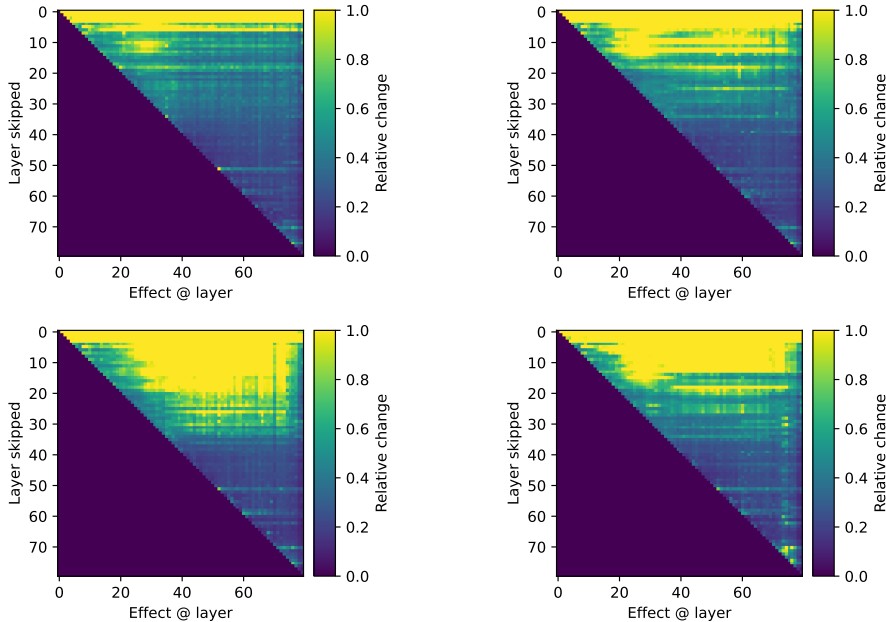

Figure 13: Individual examples for effects of skipping a layer on later layers' contribution in future timesteps (4 individual elements contributing to Fig. 3b). The variability of the importance of individual layers motivates our approach of taking the maximum over them to quantify their overall importance.

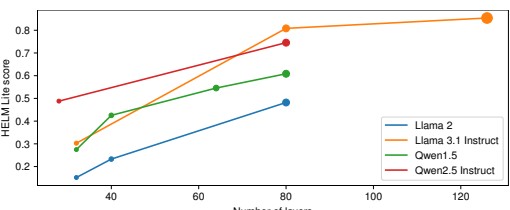

Figure 14: HELM Lite score in function of layers. The area of the dots is proportional to the parameter count. Deeper models generally perform better.

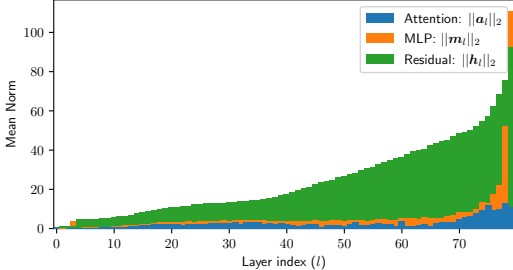

Figure 15: L2 norm of layer inputs and the sublayer contributions before summing into the residual for Llama 3.1 70B. Norm for each layer.

Prior work also studies the mechanisms that are used to perform certain operations in Transformers. Perhaps the most well-known are the induction heads [11]. In follow-up work, successor heads [45] and copy suppression [46] were discovered. Recently Lindsey et al. [12] described a large variety of circuits performing various functions in the network, including the mechanisms responsible for addition. Although these mechanisms necessarily span multiple layers, a common pattern is that they compose low-level sub-operations into a high-level operation. To the best of our knowledge,

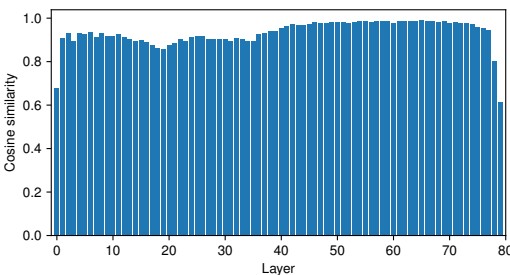

Figure 16: Cosine similarity between the residual at neighboring layers (cossim($\boldsymbol{h}_l, \boldsymbol{h}_{l+1}$)) of Llama 3.1 70B. The representations are focused on the right subspace, resulting in high cosine similarities. However, focusing on the layers' and sublayers' contributions is more informative (Fig. 2b).

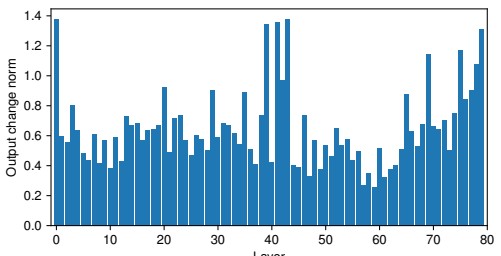

(a) Effect of skipping a layer on all predictions.

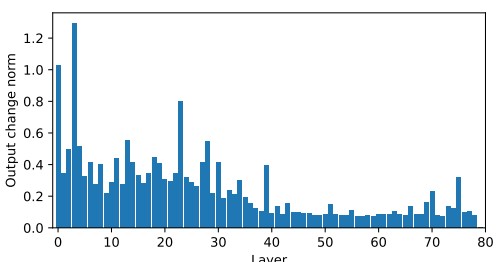

(b) Effect of skipping a layer on future predictions.

Figure 17: Analyzing the importance of layers on output predictions Llama 3.1 70B on GSM8K. The figure shows the maximum change of the output probabilities. (a) shows all tokens in the sequence when skipping a layer, including the *current* predictions, while (b) isolates the effect just for the maximum of the *future* tokens. (a) shows that despite the low effects of the layers on consecutive computations in the second half of the network (Fig. 3), the layers play an important role in the predictions. However, as (b) shows, their importance is minimal for future predictions. The second half of the layers seems to perform mostly independent, but important, computations to refine the current predicted probability distribution. This is further support for the findings of Fig. 4 in the main text.

there is no evidence of higher-level conditional composition, where a mechanism is sometimes used to directly produce the output, while other times it is used in composition with another high-level mechanism to compute a more complex function.

Previously, some methods were proposed that might help increase the effectiveness of the feature building in deeper layers. For example, Kim et al. [47] show that their method increases the angular distance between the representations of different layers, indicating that it might use deeper layers more efficiently. Li et al. [48] mix post-layernorms in early layers and pre-layernorms in later layers to improve gradient propagation. However, we are not aware of freely available popular large language models using these configurations in order to verify their effectiveness. However, we tested the OLMo 2 series of models with a different layernorm structure, resulting in similar conclusions as the standard pre-layernorm models.

The collapse of importance of attention in the later layers could be related to massive activations [49] that have been shown to collapse the attention to a few discrete tokens, potentially reducing their effectiveness. It might also partially explain the success of selective attenuation pruning [50].

More broadly, causal intervention methods gained popularity in recent years [51, 52, 53, 54, 55]. These methods are capable of providing deep insights on how neural networks operate. By direct interventions on the hypothesized mechanisms, they provide strong evidence and avoid accidental reliance on surface correlations.

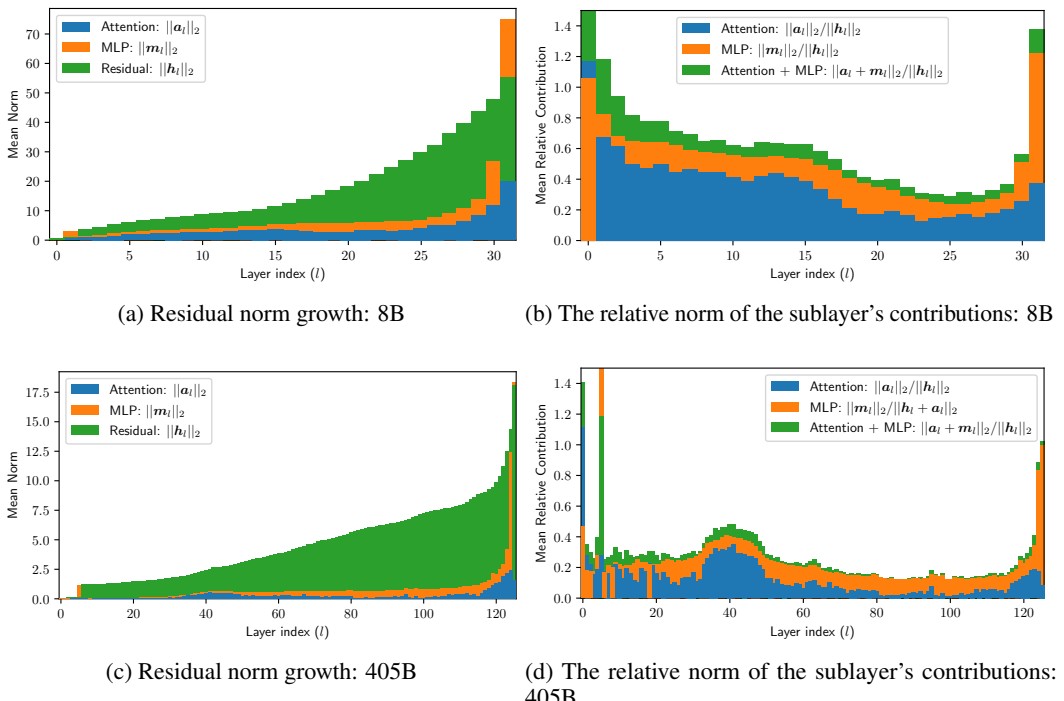

(a) Residual norm growth: 8B

(b) The relative norm of the sublayer's contributions: 8B

(c) Residual norm growth: 405B

(d) The relative norm of the sublayer's contributions: 405B

Figure 18: L2 norm of layer inputs and the sublayer contributions before summing into the residual for Llama 3.1 8B and 405B on GSM8K. (a,c) shows the norm for each layer/sublayer, while (b,d) compares the norm of the sublayer outputs to their input, quantifying the relative change induced by the sublayer (limited to max 1.5 for better visibility). A sharp drop is visible near the middle of the network. The second half of the layers changes the residual significantly less than the first half, with the exception of the last few layers. The findings are similar to what we have demonstrated in Fig. 15 and 2a.

# G    Hardware Resources

Most of our experiments were done using NNSIGHT and NDIF [21], not requiring local hardware. The experiments on the Qwen models and the Llama 3.1 70B Instruct models, which are not available on NDIF, are done on 4 Nvidia A6000 48Gb GPUs, with a rough duration of a day for the 70B experiment, and another day for all the Qwen experiments.

For Sec. 3.5, we trained each model on 2 Nvidia A100 80Gb GPUs for 2 days.

Full-finetuning Llama 3.1 3B on the DeepMind Math Dataset (Sec. D.5) was done on 4 Nvidia H200 GPUs for 10 hours.

Training the linear maps between the pair of layers of the Qwen models (Sec. 3.4) was done on A6000 GPUs, taking 80 GPU-days in total.

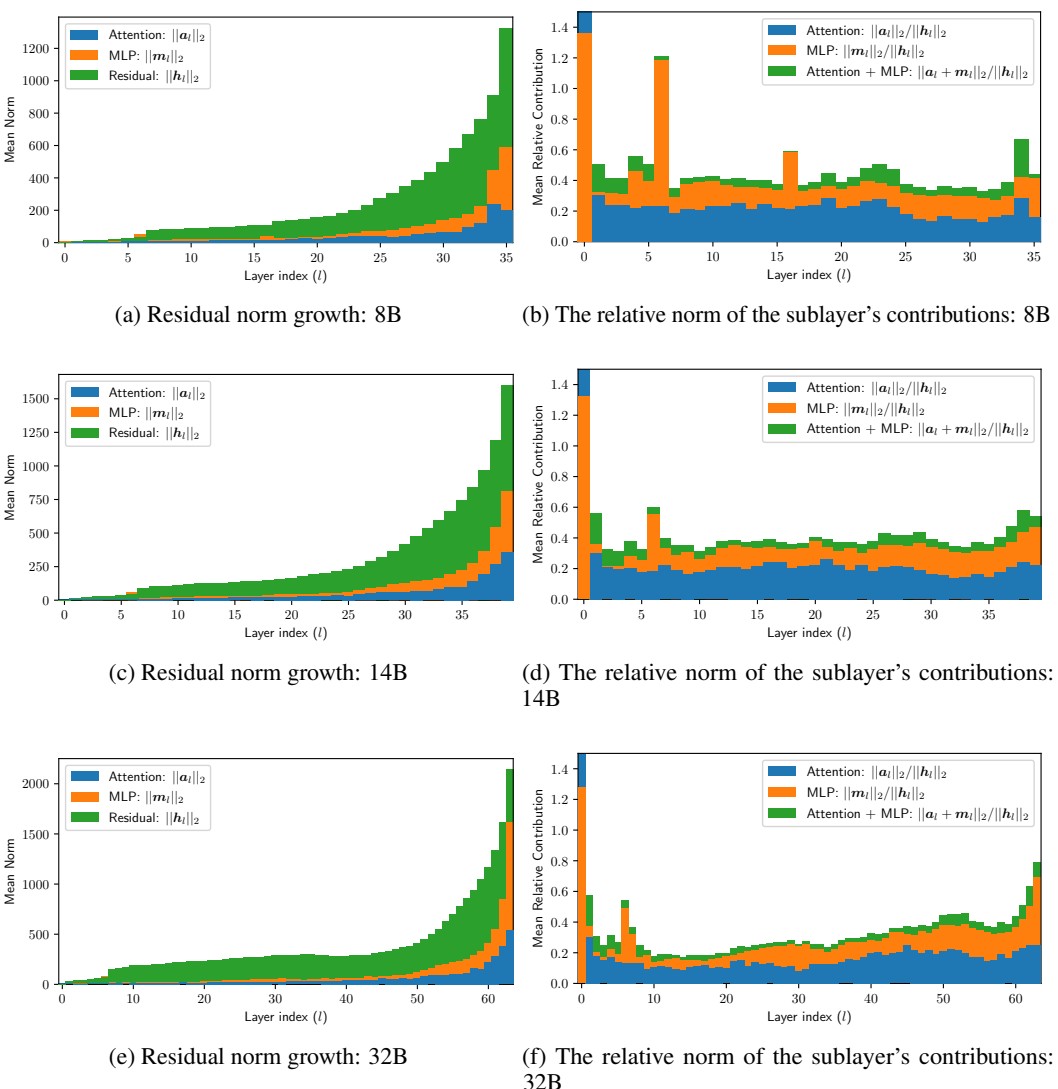

(a) Residual norm growth: 8B

(b) The relative norm of the sublayer's contributions: 8B

(c) Residual norm growth: 14B

(d) The relative norm of the sublayer's contributions: 14B

(e) Residual norm growth: 32B

(f) The relative norm of the sublayer's contributions: 32B

Figure 19: L2 norm of layer inputs and the sublayer outputs before summing into the residual for the Qwen 3 series of models on GSM8K. (a,c) shows the norm for each layer, while (b,d) compares the norm of the sublayer outputs to their input, quantifying the relative change induced by the sublayer. Relative contributions clipped to 1.5 maximum. The contribution of later layers remains more stable than the Llama models (Fig. 15 and 2a), especially for the 14B model. (e,f) Findings for Qwen 32B. This model seems to differ from all the others examined: it uses all its layers. However, the importance of the early layers is lower than the late ones.

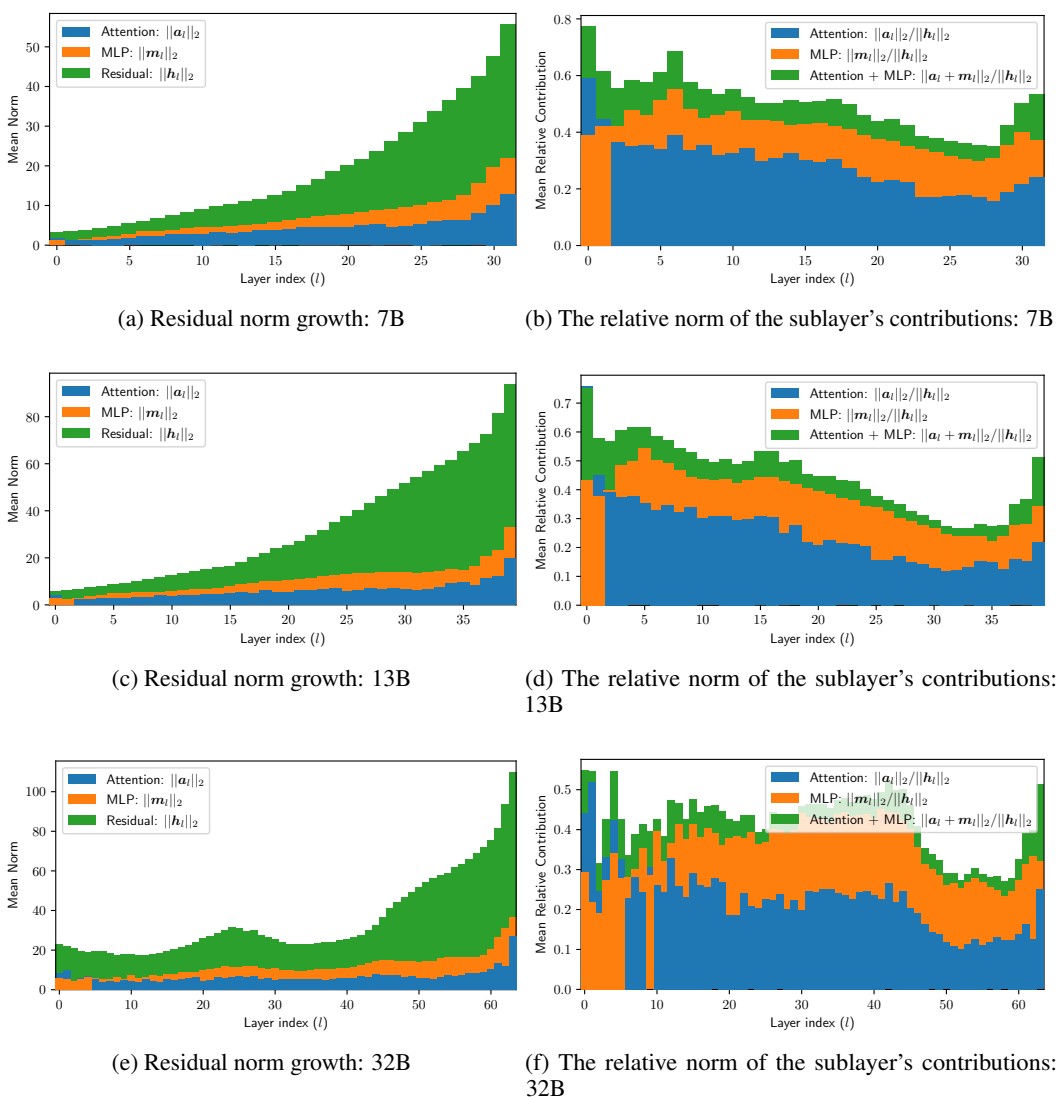

(a) Residual norm growth: 7B

(b) The relative norm of the sublayer's contributions: 7B

(c) Residual norm growth: 13B

(d) The relative norm of the sublayer's contributions: 13B

(e) Residual norm growth: 32B

(f) The relative norm of the sublayer's contributions: 32B

Figure 20: L2 norm of layer inputs and the sublayer outputs before summing into the residual for the OLMo 2 series of models on GSM8K. (a,c) shows the norm for each layer, while (b,d) compares the norm of the sublayer outputs to their input, quantifying the relative change induced by the sublayer. The relative cointribution of inidividual layers is significantly lower than the Llama (Fig. 15, 2a) and Qwen (Fig. 19) models.

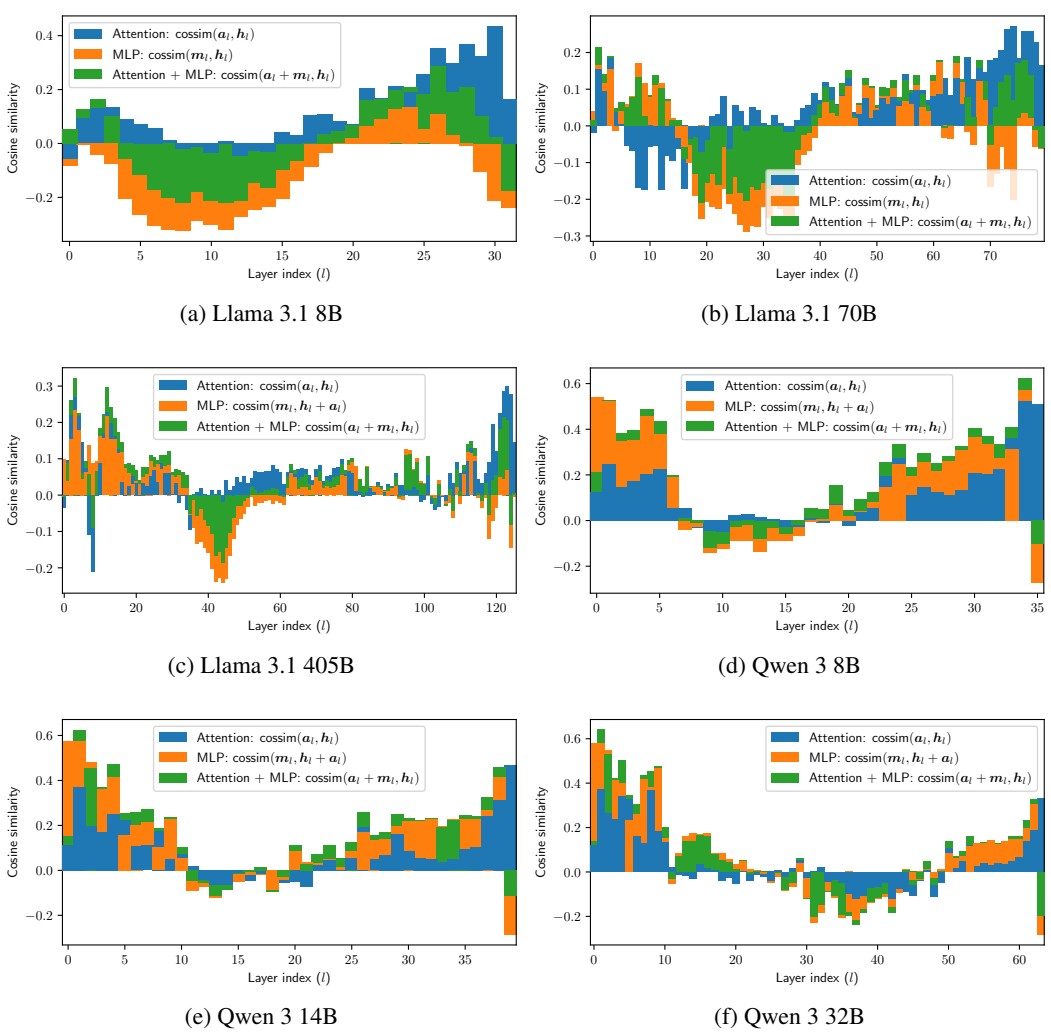

Figure 21: Cosine similarity between the sublayers' contributions and the residual for the LLama 3.1 and Qwen 3. They all show a consistent picture with Fig. 2b.

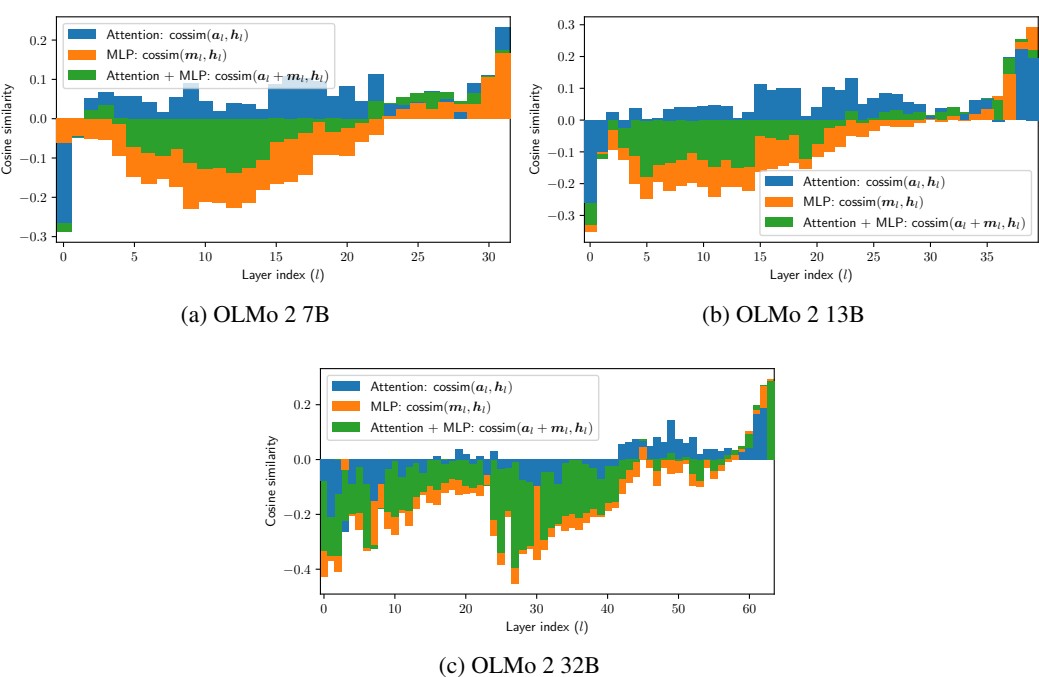

(a) OLMo 2 7B

(b) OLMo 2 13B

(c) OLMo 2 32B

Figure 22: Cosine similarity between the sublayers' contributions and the residual for OLMo 2. They all show a consistent picture with Fig. 2b.

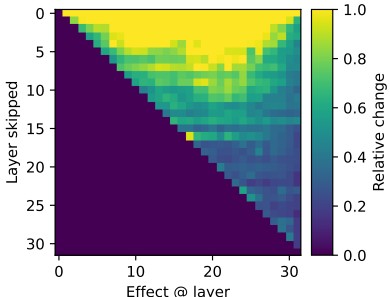

(a) Effect of skipping a layer on later layers in future timesteps: 8B

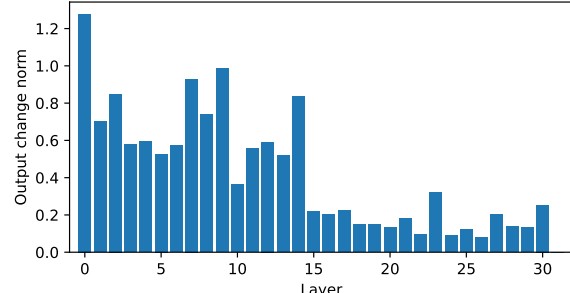

(b) Effect of skipping a layer on future predictions: 8B

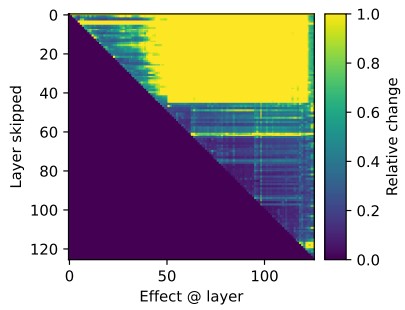

(c) Effect of skipping a layer on later layers in future timesteps: 405B

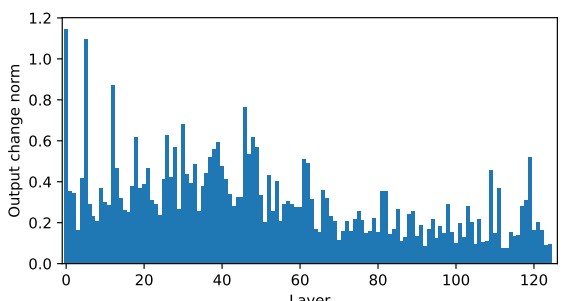

(d) Effect of skipping a layer on future predictions: 405B

Figure 23: Analyzing the importance of layers on computations in later layers and output predictions for Llama 3.1 8B and 405B on GSM8K, focusing on the effect on future tokens. (a,c) The maximum relative change in the layer's output when a previous layer is skipped. The second half of the layers has a weaker effect on future computations compared to the first. The range is limited between 0 and 1. (b,d) The maximum change in the output probabilities. The findings are identical to the ones discussed in Fig. 3.

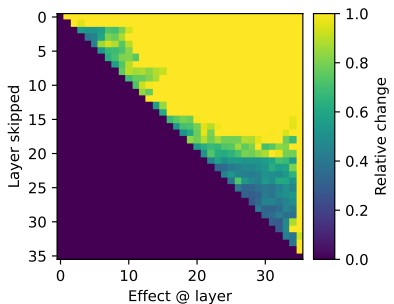

(a) Effect of skipping a layer on later layers in future timesteps: 8B

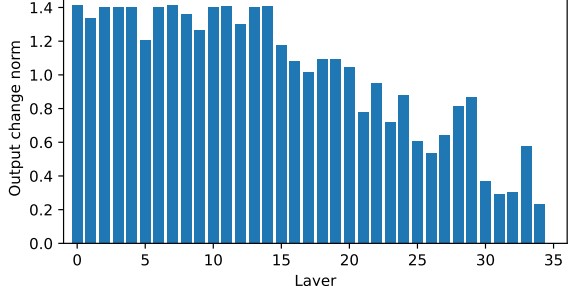

(b) Effect of skipping a layer on future predictions: 8B

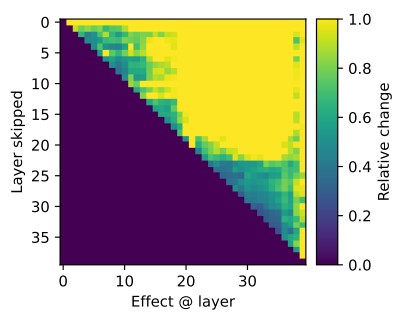

(c) Effect of skipping a layer on later layers in future timesteps: 14B

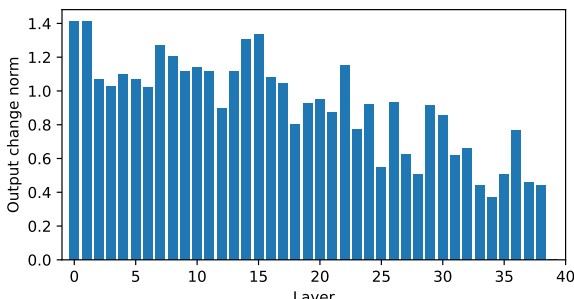

(d) Effect of skipping a layer on future predictions: 14B

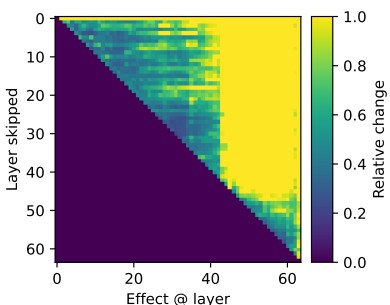

(e) Effect of skipping a layer on later layers in future timesteps: 32B

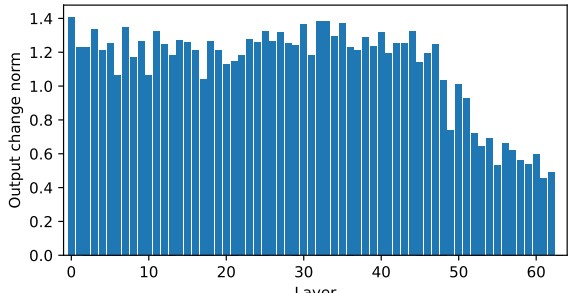

(f) Effect of skipping a layer on future predictions: 32B

Figure 24: Analyzing the importance of layers on computations in later layers and output predictions on Qwen 3 series of models on GSM8K, focusing on the effect on future tokens. (a,c,e) The maximum relative change in the layer's output when a previous layer is skipped. The second half of the layers has a weaker effect on future computations compared to the first. The range is limited between 0 and 1. (b,d,f) The maximum change in the output probabilities. The findings for the 8 and 14B models are identical to the ones discussed in Fig. 3. However, the 32B model behaves differently: it also displays the reduced effects on future predictions in the late layers, but more interestingly, the lower layers seem not to build on each other's computation, but just accumulate information in the residual, which will be used in late layers.

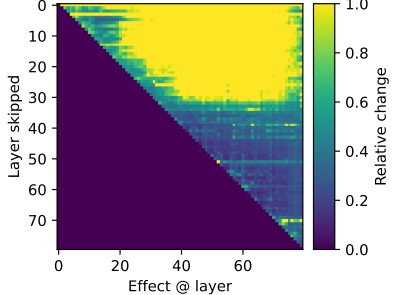 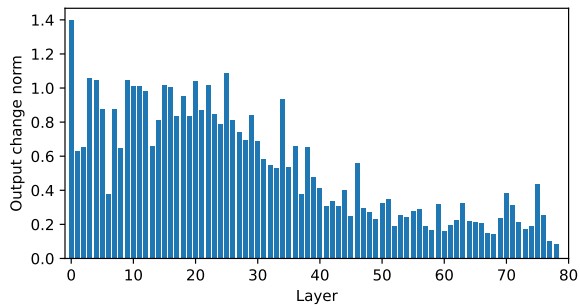

(a) Effect of skipping a layer on later layers in future timesteps.

(b) Effect of skipping a layer on future predictions.

Figure 25: Analyzing layer importance for future predictions in Llama 3.1 70B Instruct. (a) The maximum relative change in the layer's output when a previous layer is skipped. It can be seen that layers in the second half of the model have minimal effect on the future computations. (b) The maximum relative change in the output probabilities. Instruction tuning seems to somewhat increase all layers' significance to the future predictions. However, the stark difference between the first and second halves of the model is still present. Compare to Fig. 3b and 17b.

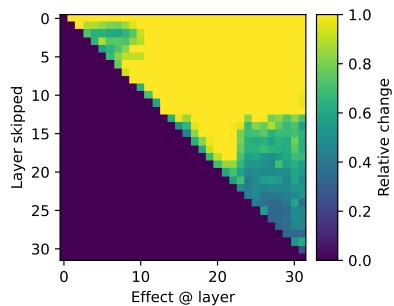

(a) Effect of skipping a layer on later layers in future timesteps: 7B

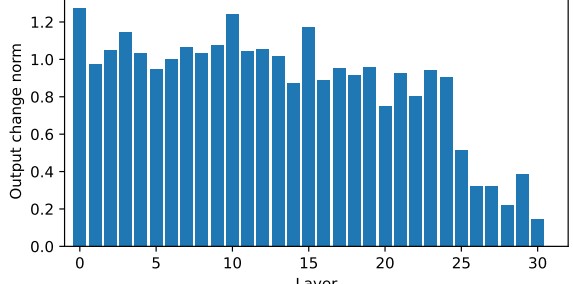

(b) Effect of skipping a layer on future predictions: 7B

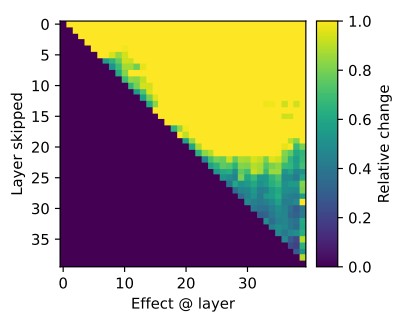

(c) Effect of skipping a layer on later layers in future timesteps: 13B

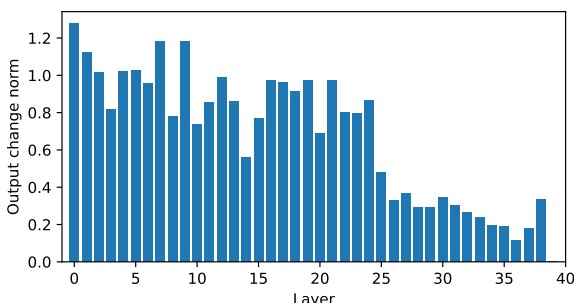

(d) Effect of skipping a layer on future predictions: 13B

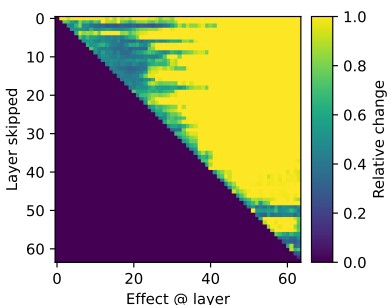

(e) Effect of skipping a layer on later layers in future timesteps: 32B

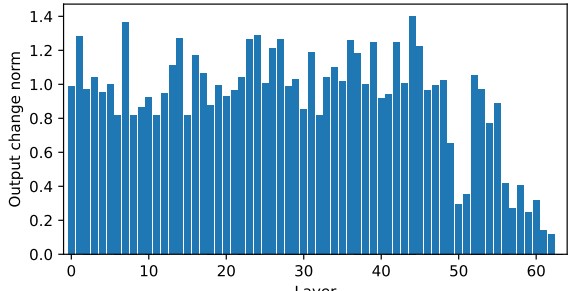

(f) Effect of skipping a layer on future predictions: 32B

Figure 26: Analyzing the importance of layers on computations in later layers and output predictions on OLMo 2 series of models on GSM8K, focusing on the effect on future tokens. (a,c,e) The maximum relative change in the layer's output when a previous layer is skipped. (a,c) The second half of the layers has a weaker effect on future computations compared to the first. The range is limited between 0 and 1. (b,d,f) The maximum change in the output probabilities. The findings for the 8 and 13B models are identical to the ones discussed in Fig. 3. However, the 32B model behaves similarly to Qwen 3 32B (Fig. 24e): it also displays the reduced effects on future predictions in the late layers, but more interestingly, the lower layers seem not to build on each other's computation, but just accumulate information in the residual, which will be used in late layers.

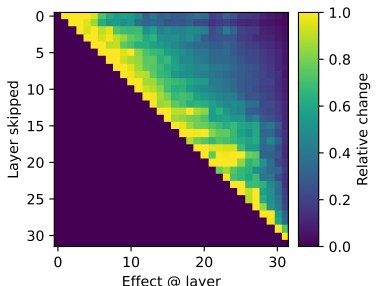

(a) Llama 3.1 8B: Local effect of layer on later layers' contributions in the *all* timesteps.

(b) Llama 3.1 8B: Local effect of layer on later layers' contributions in *future* timesteps.

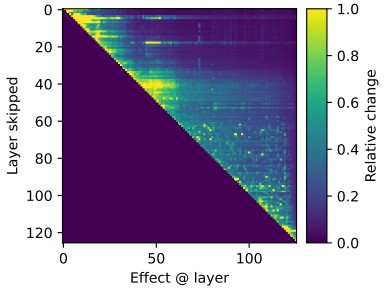

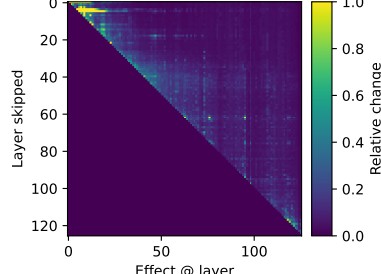

(c) Llama 3.1 405B: Local effect of layer on later layers' contributions in the *all* timesteps.

(d) Llama 3.1 405B: Local effect of layer on later layers' contributions in *future* timesteps.

Figure 27: Analyzing the direct local effects between pairs of layers of Llama 3.1 models. It highlights layer pairs with a direct effect on each other. The effects are not propagated to future layers. For each layer $s$, the plot shows future layers that build on the representation computed by $s$. (a,c) Effects on all tokens, highlighting all possible circuits. (b,d) Effect on future tokens. The sparse, bright spots indicate multi-layer, multi-token mechanisms, such as induction heads. Note that interacting layers are not necessarily spatially close to each other.

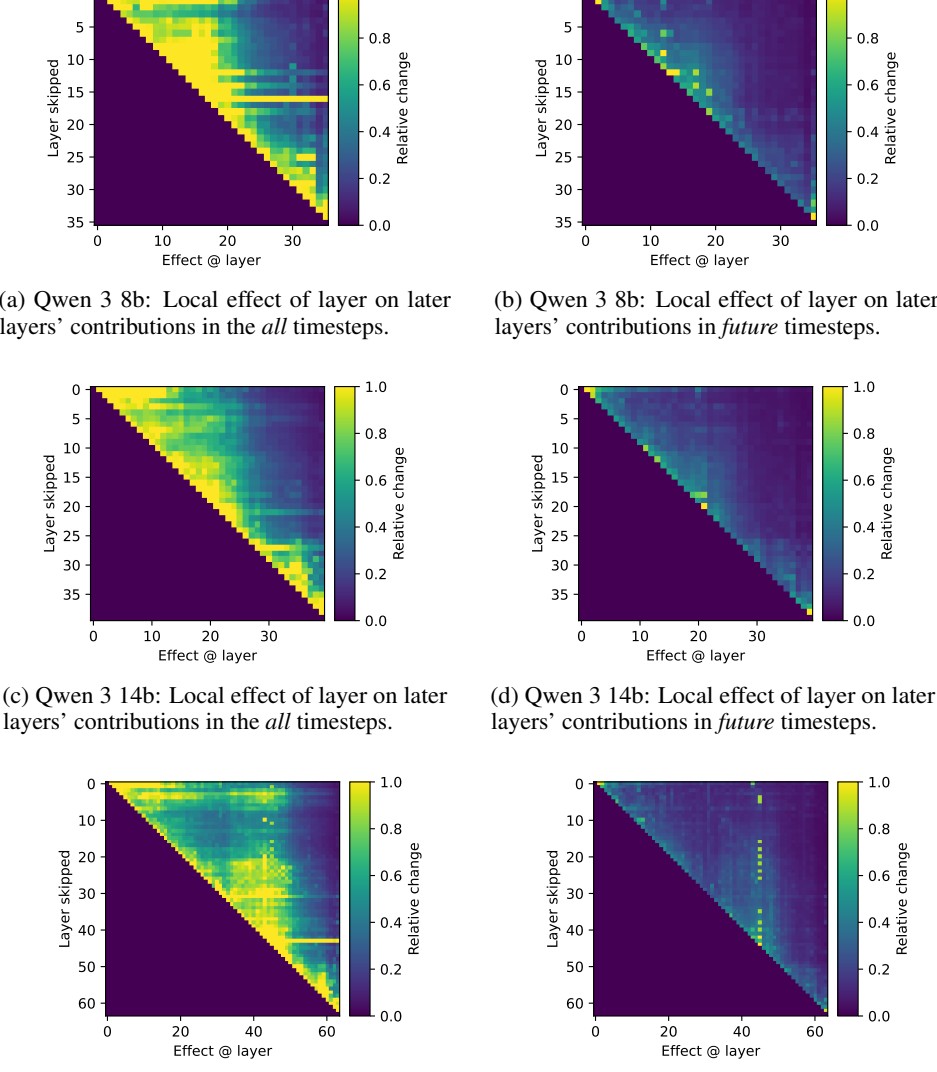

(a) Qwen 3 8b: Local effect of layer on later layers' contributions in the *all* timesteps.

(b) Qwen 3 8b: Local effect of layer on later layers' contributions in *future* timesteps.

(c) Qwen 3 14b: Local effect of layer on later layers' contributions in the *all* timesteps.

(d) Qwen 3 14b: Local effect of layer on later layers' contributions in *future* timesteps.

(e) Qwen 3 32b: Local effect of layer on later layers' contributions in the *all* timesteps.

(f) Qwen 3 32b: Local effect of layer on later layers' contributions in *future* timesteps.

Figure 28: Analyzing the direct local effects between pairs of layers of Qwen models. It highlights layer pairs with a direct effect on each other. The effects are not propagated to future layers. For each layer $s$, the plot shows future layers that build on the representation computed by $s$. (a,c,e) Effects on all tokens, highlighting all possible circuits. (b,d,f) Effect on future tokens. The sparse, bright spots indicate multi-layer, multi-token mechanisms, such as induction heads. Note that interacting layers are not necessarily spatially close to each other. Interestingly, Qwen 3 32b shows a single layer that moves most of the features at once from previous layers to future tokens.

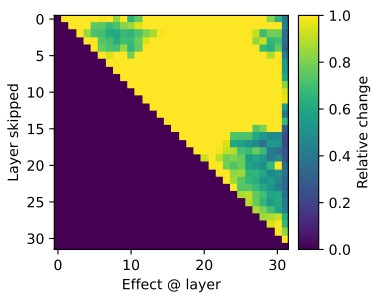

(a) OLMo 2 7b: Local effect of layer on later layers' contributions in the *all* timesteps.

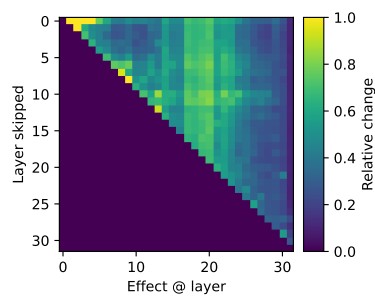

(b) OLMo 2 7b: Local effect of layer on later layers' contributions in *future* timesteps.

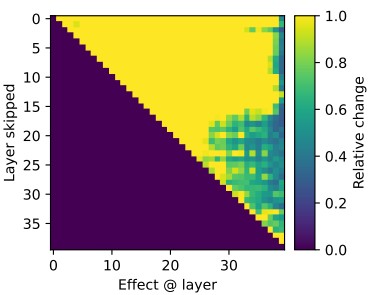

(c) OLMo 2 13b: Local effect of layer on later layers' contributions in the *all* timesteps.

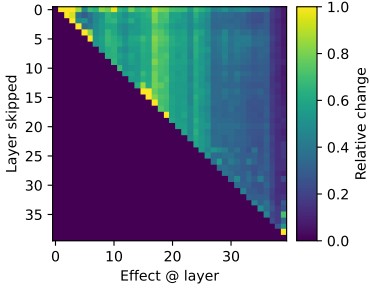

(d) OLMo 2 13b: Local effect of layer on later layers' contributions in *future* timesteps.

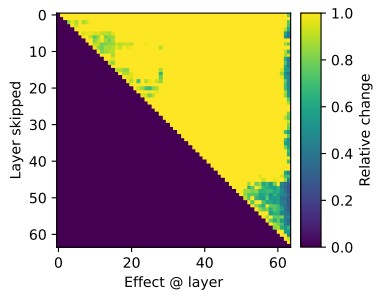

(e) OLMo 2 32b: Local effect of layer on later layers' contributions in the *all* timesteps.

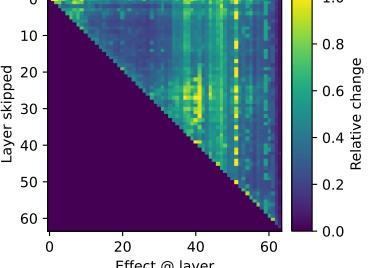

(f) OLMo 2 32b: Local effect of layer on later layers' contributions in *future* timesteps.

Figure 29: Analyzing the direct local effects between pairs of layers of OLMo 2 models. It highlights layer pairs with a direct effect on each other. The effects are not propagated to future layers. For each layer $s$, the plot shows future layers that build on the representation computed by $s$. (a,c,e) Effects on all tokens, highlighting all possible circuits. (b,d,f) Effect on future tokens. The OLMo models seem to have significantly stronger contributions to both the same token (a,c,e) and the future tokens (b,d,f), compared to the LLama (Fig. 5, 27) and Qwen (Fig. 28) models. This can be probably attributed to their reordered norm, where the normalization is applied after the layers, before merging back to the residual.

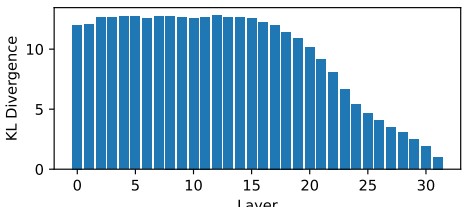

(a) Llama 3.1 8B: KL divergence between Log-itlens and final prediction

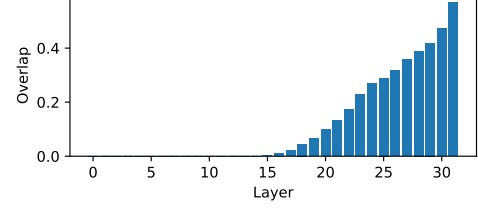

(b) Llama 3.1 8B: Overlap in top-5 predicted tokens

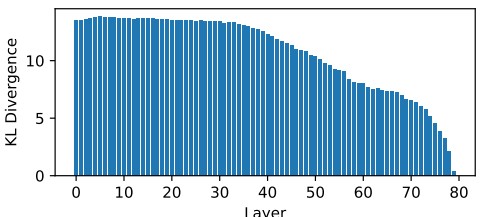

(c) Llama 3.1 70B Instruct: KL divergence between Logitlens and final prediction

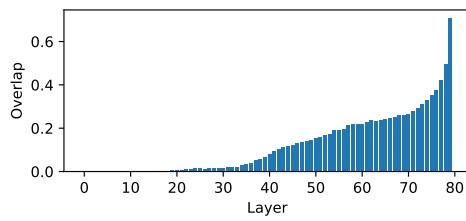

(d) Llama 3.1 70B Instruct: Overlap in top-5 predicted tokens

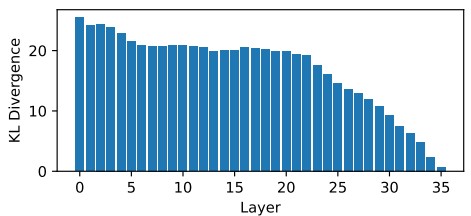

(e) Qwen 3 8B: KL divergence between Logitlens and final prediction

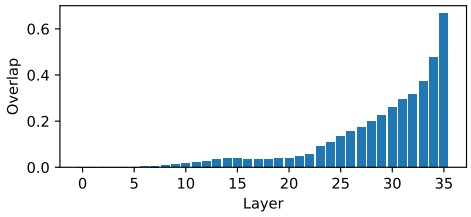

(f) Qwen 3 8B: Overlap in top-5 predicted tokens

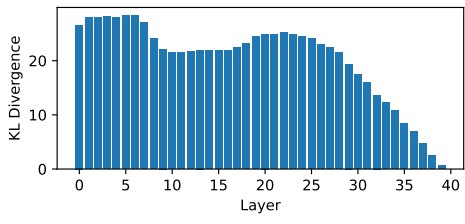

(g) Qwen 3 14B: KL divergence between Logitlens and final prediction

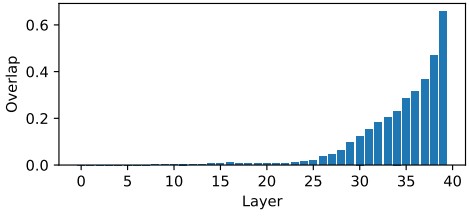

(h) Qwen 3 14B: Overlap in top-5 predicted tokens

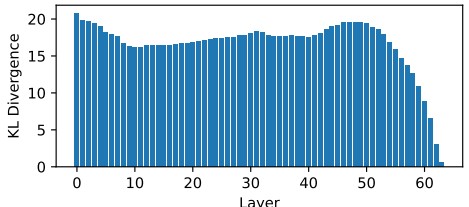

(i) Qwen 3 32B: KL divergence between Logitlens and final prediction

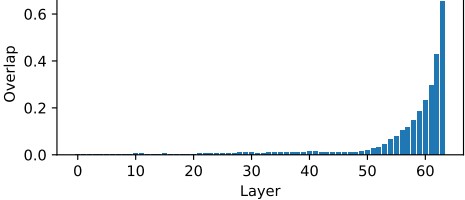

(j) Qwen 3 32B: Overlap in top-5 predicted tokens

Figure 30: Comparing Logitlens probes from different layers to the final prediction for different models. Left: KL divergence between the output of the Logitlens and the final prediction. Right: Overlap between the top-5 tokens predicted by Logitlens and the final model prediction.

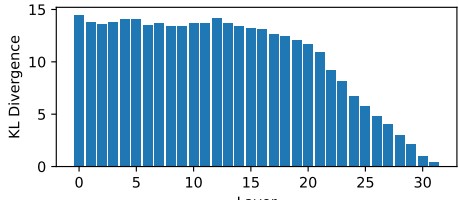

(a) OLMo 2 7B: KL divergence between Logitlens and final prediction

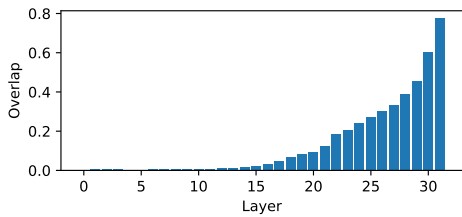

(b) OLMo 2 7B: Overlap in top-5 predicted tokens

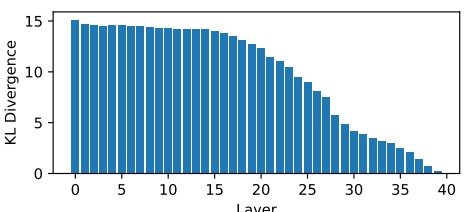

(c) OLMo 2 13B: KL divergence between Logitlens and final prediction

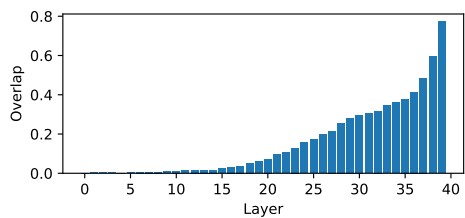

(d) OLMo 2 13B: Overlap in top-5 predicted tokens

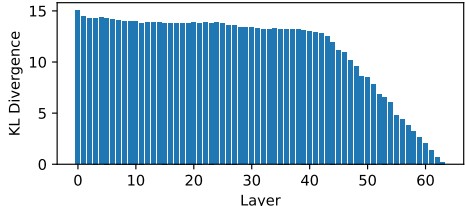

(e) OLMo 2 32B: KL divergence between Logitlens and final prediction

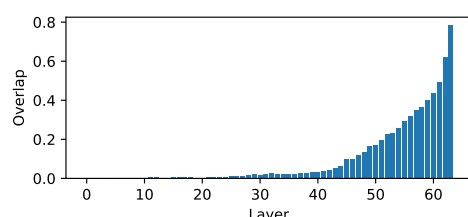

(f) OLMo 2 32B: Overlap in top-5 predicted tokens

Figure 31: Comparing Logitlens probes from different layers to the final prediction for OLMo 2 models. Left: KL divergence between the output of the Logitlens and the final prediction. Right: Overlap between the top-5 tokens predicted by Logitlens and the final model prediction.

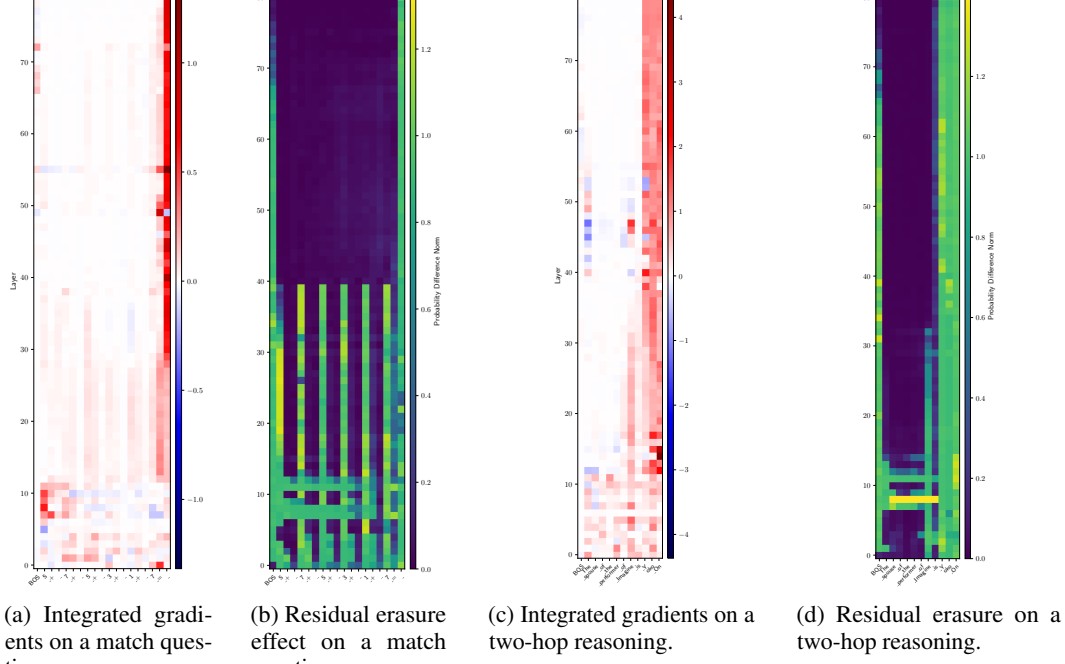

(a) Integrated gradients on a match question.

(b) Residual erasure effect on a match question.

(c) Integrated gradients on a two-hop reasoning.

(d) Residual erasure on a two-hop reasoning.

Figure 32: Analyzing the effect of individual computation steps on Llama 3.1 70B. (a,b) Basic math question. (c,d) Two-hop reasoning. Note that the answer is 4 tokens long in this case, providing a stronger gradient signal. (a,c) Integrated gradients. (b,d) The probability distribution change ($\|y - \bar{y}\|_2$) when erasing the residual of a given token in a given layer. This score shows until when the information from a column is used. In both cases, the second half of the model shows minimal effect. Moreover, in arithmetic, later hops of computation do not use more depth, indicating that no composition is happening.

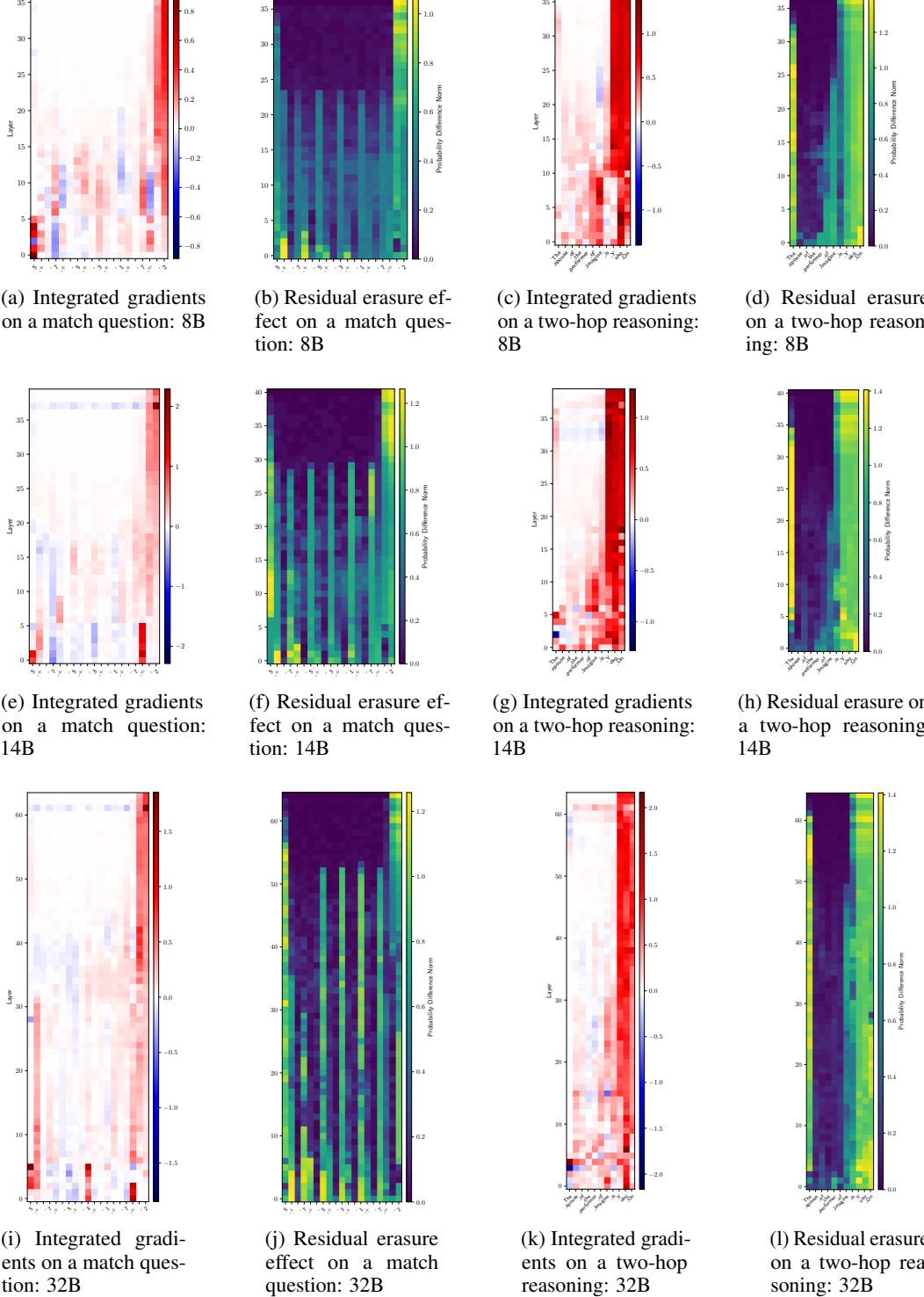

(a) Integrated gradients on a match question: 8B

(b) Residual erasure effect on a match question: 8B

(c) Integrated gradients on a two-hop reasoning: 8B

(d) Residual erasure on a two-hop reasoning: 8B

(e) Integrated gradients on a match question: 14B

(f) Residual erasure effect on a match question: 14B

(g) Integrated gradients on a two-hop reasoning: 14B

(h) Residual erasure on a two-hop reasoning: 14B

(i) Integrated gradients on a match question: 32B

(j) Residual erasure effect on a match question: 32B

(k) Integrated gradients on a two-hop reasoning: 32B

(l) Residual erasure on a two-hop reasoning: 32B

Figure 33: Analyzing the effect of individual computation steps on the Qwen 3 series of models. (a,b,e,f,i,j) Basic math question. (c,d,g,h,k,l) Two-hop reasoning. Note that the answer is 4 tokens long in this case, providing a stronger gradient signal. (a,c,e,g,i,k) Integrated gradients. (b,d,f,h,j,l) The probability distribution change ($||\boldsymbol{y} - \bar{\boldsymbol{y}}||_2$) when erasing the residual of a given token in a given layer. This score shows until when the information from a column is used. The models use more layers compared to the Llama series (Fig. 6), but still show no increased depth for later computations.

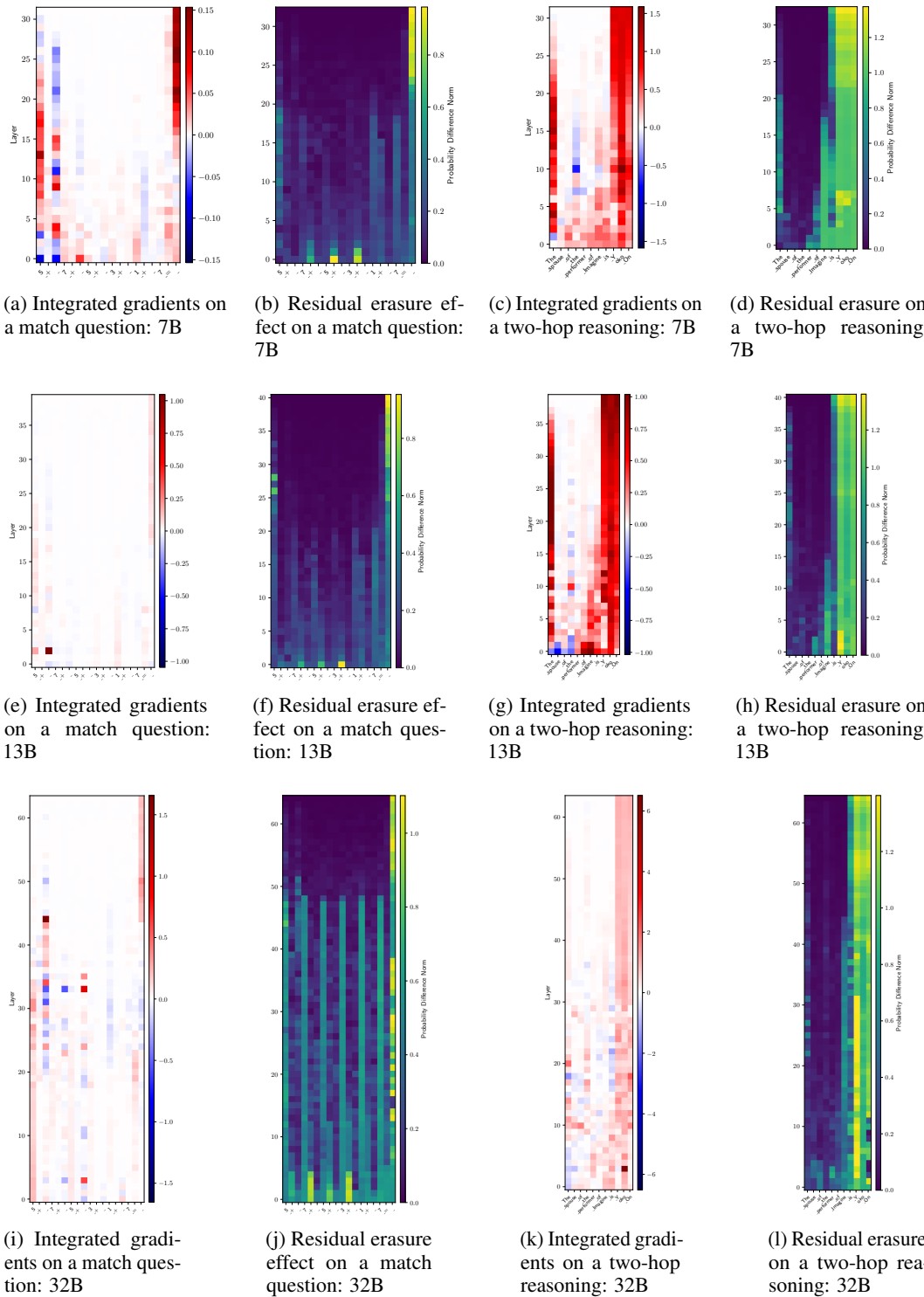

(a) Integrated gradients on a match question: 7B

(b) Residual erasure effect on a match question: 7B

(c) Integrated gradients on a two-hop reasoning: 7B

(d) Residual erasure on a two-hop reasoning: 7B

(e) Integrated gradients on a match question: 13B

(f) Residual erasure effect on a match question: 13B

(g) Integrated gradients on a two-hop reasoning: 13B

(h) Residual erasure on a two-hop reasoning: 13B

(i) Integrated gradients on a match question: 32B

(j) Residual erasure effect on a match question: 32B

(k) Integrated gradients on a two-hop reasoning: 32B

(l) Residual erasure on a two-hop reasoning: 32B

Figure 34: Analyzing the effect of individual computation steps on the OLMo 2 series of models. (a,b,e,f,i,j) Basic math question. (c,d,g,h,k,l) Two-hop reasoning. Note that the answer is 4 tokens long in this case, providing a stronger gradient signal. (a,c,e,g,i,k) Integrated gradients. (b,d,f,h,j,l) The probability distribution change ($||\boldsymbol{y} - \bar{\boldsymbol{y}}||_2$) when erasing the residual of a given token in a given layer. This score shows until when the information from a column is used. Up to 13B, the models seems to do shallower processing than both the Llama series (Fig. 6) and Qwen 3 (Fig. 33).

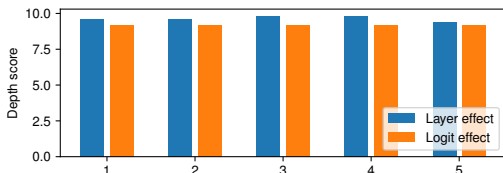

(a) Depth score on MATH dataset on different difficulty examples: 8B

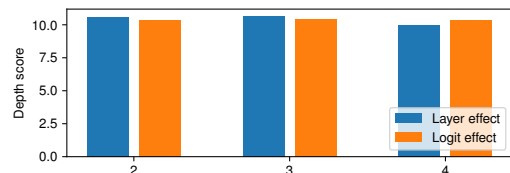

(b) Depth score on MQuAKE on different number of hops: 8B

Figure 35: Depth score for Llama 8B: the weighted average of layer index with its importance, as a function of a given difficulty metric. Importance is measured based on both the effect on future internal computations and on the effect on future predictions. (a) MATH dataset. The x-axis is the difficulty level defined by the dataset. (b) MQuAKE. The x-axis is the number of hops in the question. The findings are similar to Llama 8B (Fig. 7).

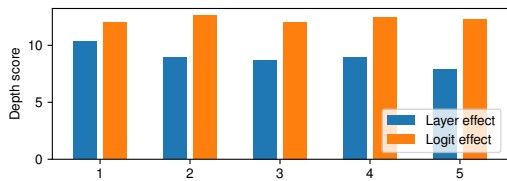

(a) Depth score on MATH dataset on different difficulty examples: 8B



(b) Depth score on MQuAKE on different number of hops: 8B

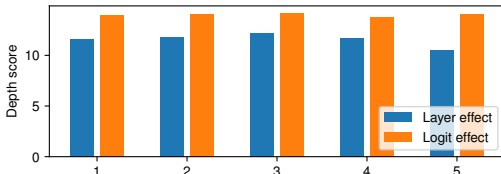

(c) Depth score on MATH dataset on different difficulty examples: 14B



(d) Depth score on MQuAKE on different number of hops: 14B

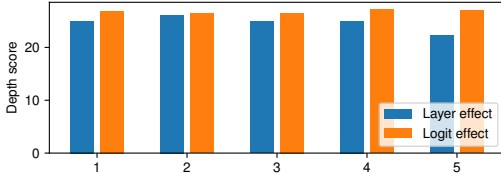

(e) Depth score on MATH dataset on different difficulty examples: 32B

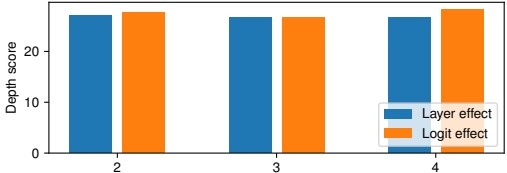

(f) Depth score on MQuAKE on different number of hops: 32B

Figure 36: Depth score for the Qwen 3 series of models: the weighted average of layer index with its importance, as a function of a given difficulty metric. Importance is measured based on both the effect on future internal computations and on the effect on future predictions. (a,c,e) MATH dataset. The x-axis is the difficulty level defined by the dataset. (b,d,f) MQuAKE. The x-axis is the number of hops in the question. The findings are identical to the Llama models. See Fig. 7.

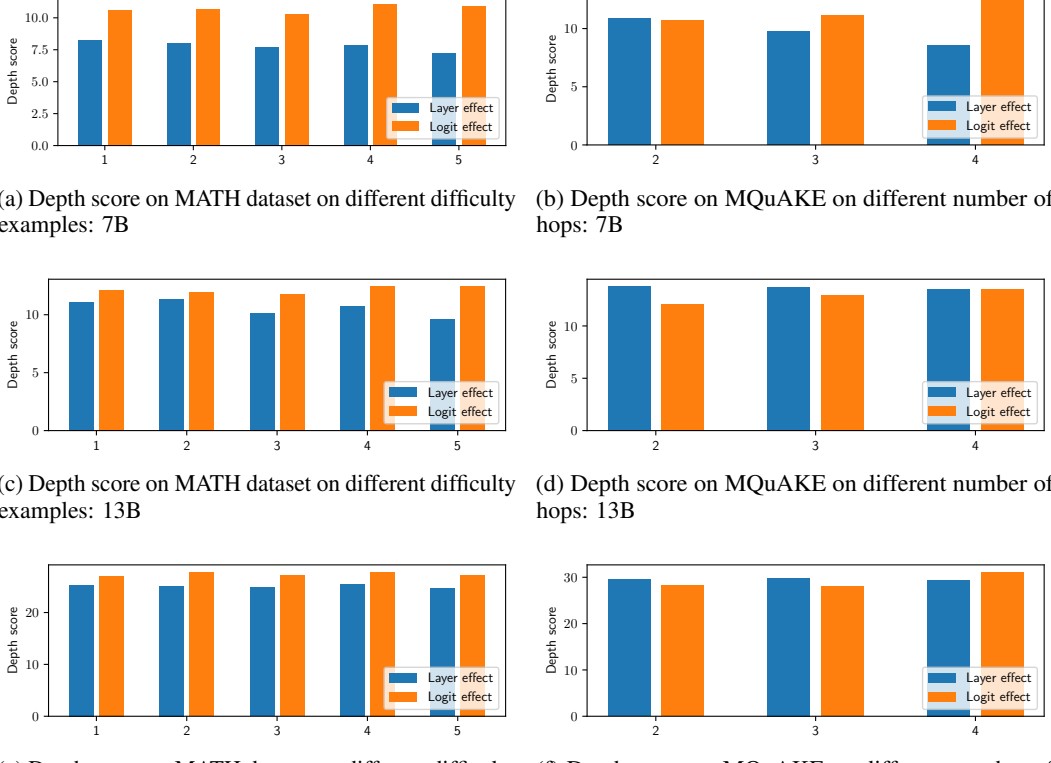

(a) Depth score on MATH dataset on different difficulty examples: 7B

(b) Depth score on MQuAKE on different number of hops: 7B

(c) Depth score on MATH dataset on different difficulty examples: 13B

(d) Depth score on MQuAKE on different number of hops: 13B

(e) Depth score on MATH dataset on different difficulty examples: 32B

(f) Depth score on MQuAKE on different number of hops: 32B

Figure 37: Depth score for the OLMo 2 series of models: the weighted average of layer index with its importance, as a function of a given difficulty metric. Importance is measured based on both the effect on future internal computations and on the effect on future predictions. (a,c,e) MATH dataset. The x-axis is the difficulty level defined by the dataset. (b,d,f) MQuAKE. The x-axis is the number of hops in the question.. The findings are very similar to the Llama and Qwen models. See Figs. 7, 36.

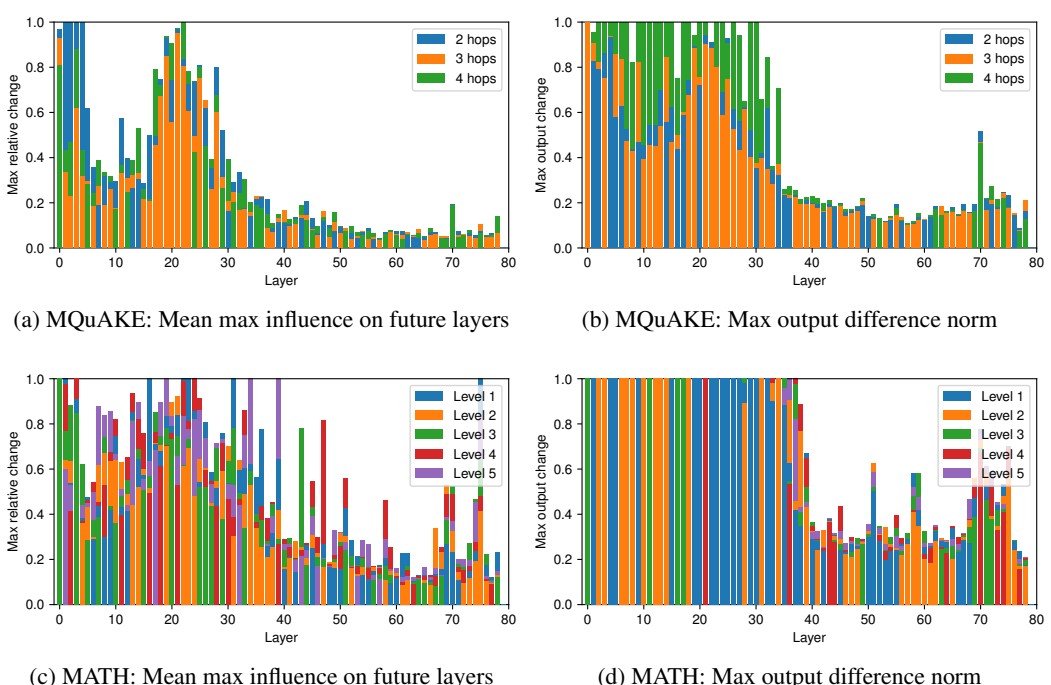

(a) MQuAKE: Mean max influence on future layers

(b) MQuAKE: Max output difference norm

(c) MATH: Mean max influence on future layers

(d) MATH: Max output difference norm

Figure 38: Layerwise effect of different complexity computations. (a,b) Questions with a different number of hops from the MQuAKE dataset. (c,d) Problems with different difficulty levels from the MATH dataset. (a,c) The max relative change in the future layer's contribution to the answer when a given layer is skipped. Mean over all future layers. (b,d) Maximum L2 norm of the change in the output probability distribution ($||\boldsymbol{y} - \bar{\boldsymbol{y}}||_2$). If more complex computations use more layers, we would expect that the importance of deeper layers increases with complexity. However, we see no evidence of such a pattern.

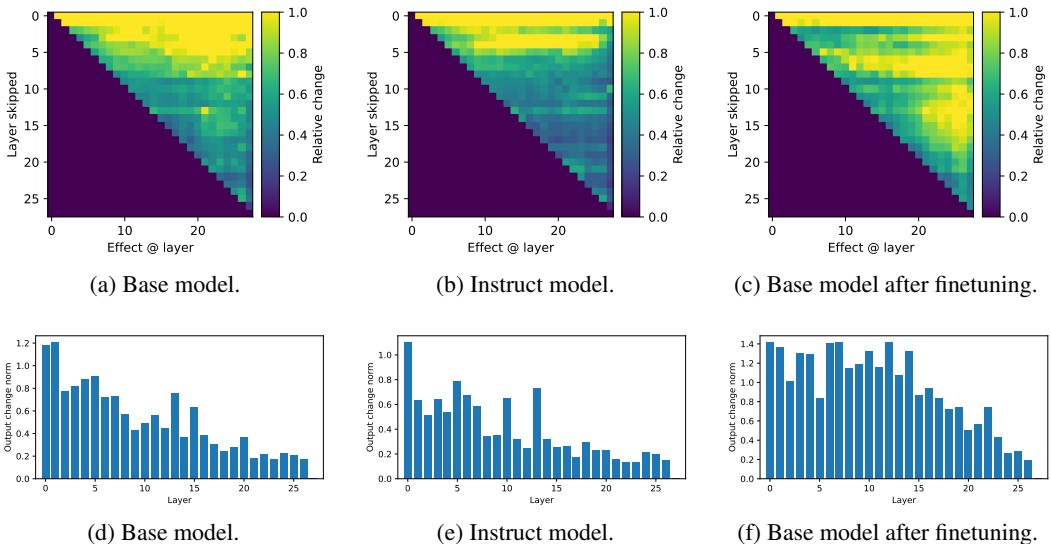

(a) Base model.  (b) Instruct model.  (c) Base model after finetuning.

(d) Base model.  (e) Instruct model.  (f) Base model after finetuning.

Figure 39: The effect of fine-tuning on the max future effects of Llama 3.2 3B on the DeepMind Math dataset's arithmetic splits. (a,b,c) Effect of skipping a layer on the later layers of future tokens. (d,e,f) Effect on future predictions. Max over 20 random examples from the validation set. The fine-tuning seems to increase the importance of the later layers at first glance. However, looking at individual examples reveals that the effect is only marginal, affecting the last 1-2 tokens before the prediction (Fig. 40).

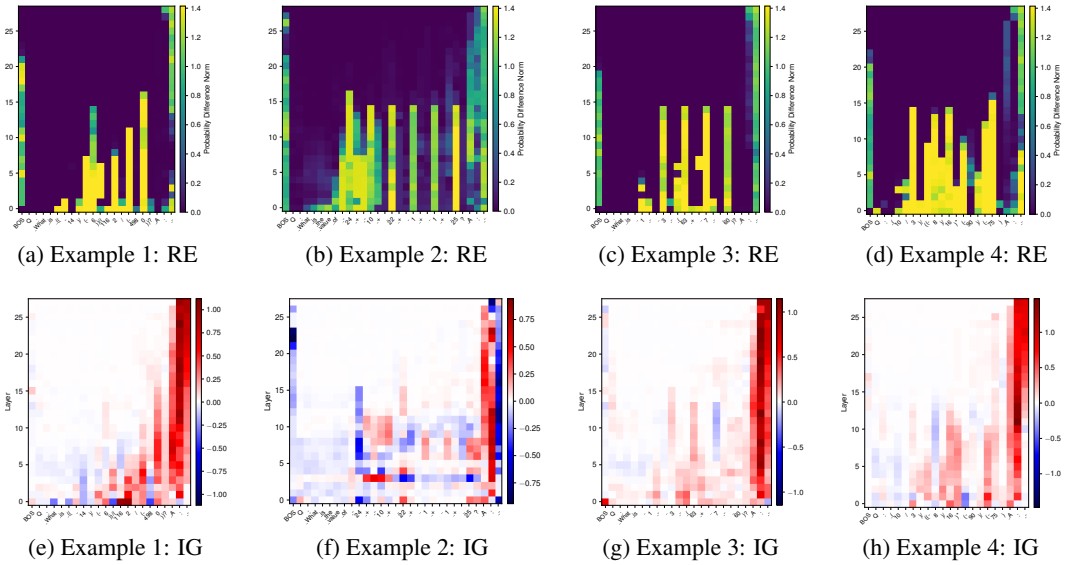

(a) Example 1: RE  (b) Example 2: RE  (c) Example 3: RE  (d) Example 4: RE

(e) Example 1: IG  (f) Example 2: IG  (g) Example 3: IG  (h) Example 4: IG

Figure 40: Analyzing residual erasure and integrated gradients on Llama 3.2 3B fine-tuned on the DeepMind Math dataset. Even though Fig. 39 indicates deeper computations compared to the base model, looking at individual examples reveals that the effect concentrates only on the last 1-2 tokens before the answer, indicating that the effect is superficial.

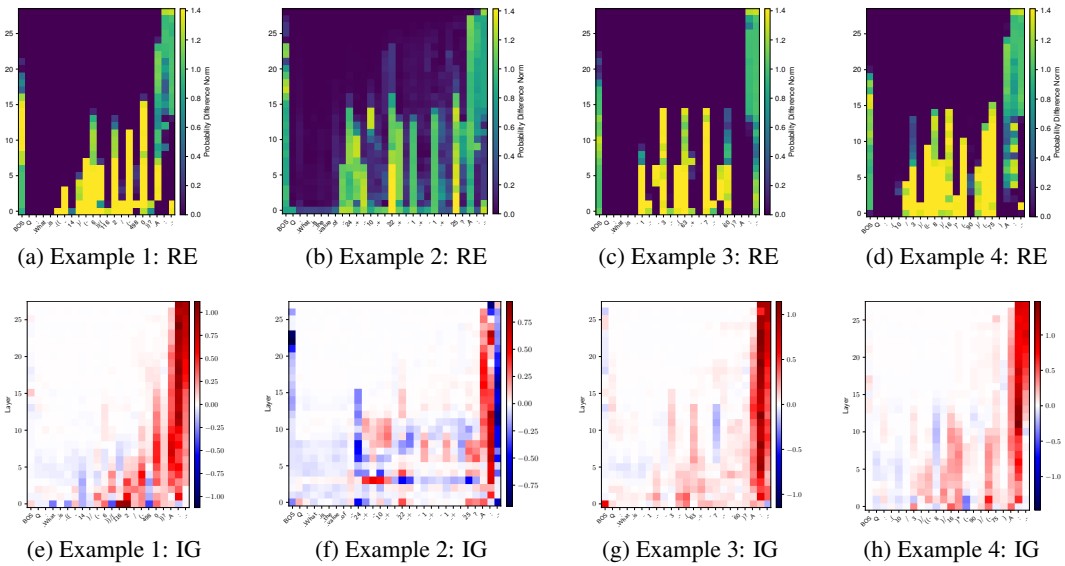

(a) Example 1: RE    (b) Example 2: RE    (c) Example 3: RE    (d) Example 4: RE

(e) Example 1: IG    (f) Example 2: IG    (g) Example 3: IG    (h) Example 4: IG

Figure 41: Analyzing residual erasure and integrated gradients on Llama 3.2 3B fine-tuned on the DeepMind Math dataset, when trained without modeling the question. Not modeling the uncertainty in the question seems not to make any difference for fine-tuning. Compare to Fig. 40.

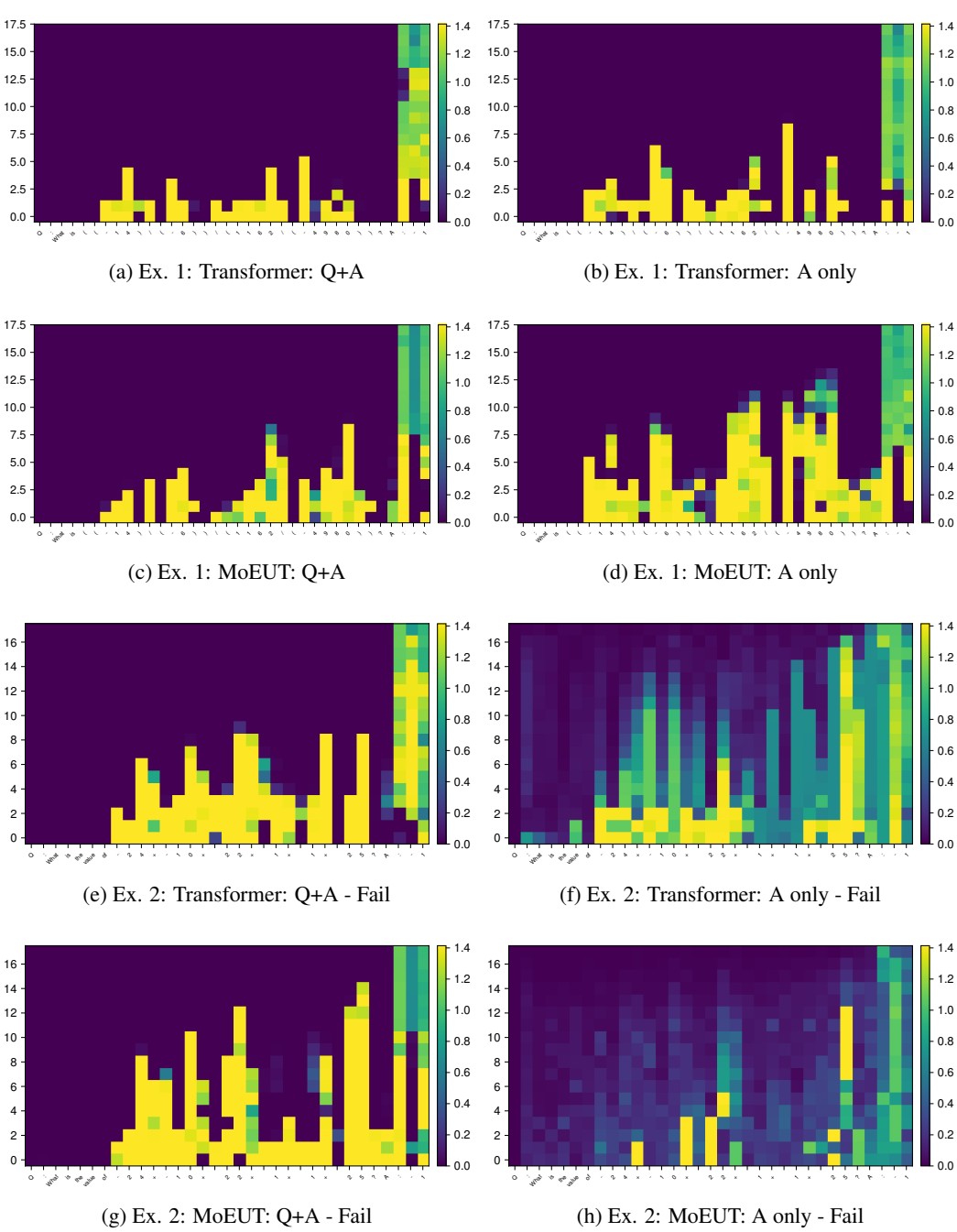

(a) Ex. 1: Transformer: Q+A

(b) Ex. 1: Transformer: A only

(c) Ex. 1: MoEUT: Q+A

(d) Ex. 1: MoEUT: A only

(e) Ex. 2: Transformer: Q+A - Fail

(f) Ex. 2: Transformer: A only - Fail

(g) Ex. 2: MoEUT: Q+A - Fail

(h) Ex. 2: MoEUT: A only - Fail

Figure 42: Training Transformer and MoEUT models from scratch on DeepMind Math dataset arithmetic subset, with and without applying loss to the question part of the input. We show the residual erasure experiments here, with identical examples to Fig. 40. (a,e) We can see that if the model is trained with the question modeling enabled, it does not use its 2nd half of the layers, similarly to the LLMs. (b,f) If modeling the question only, the model uses significantly more layers. (c,d,g,h) MoEUT successfully uses more layers even when modeling the question, although modeling the answer only seem to help further (f). All models failed to answer Example 2 correctly (e,f,g,h). Fig. 43.

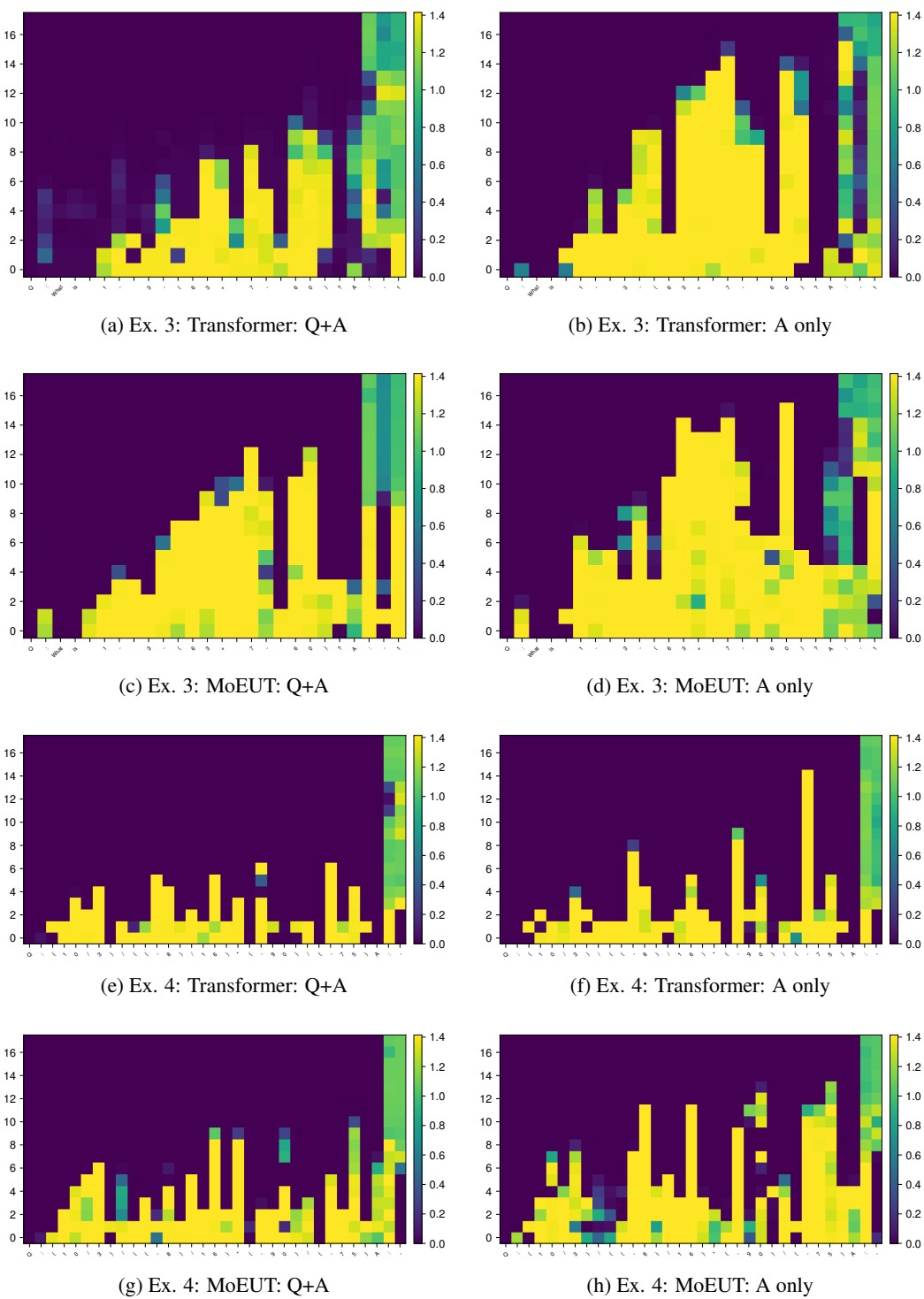

(a) Ex. 3: Transformer: Q+A

(b) Ex. 3: Transformer: A only

(c) Ex. 3: MoEUT: Q+A

(d) Ex. 3: MoEUT: A only

(e) Ex. 4: Transformer: Q+A

(f) Ex. 4: Transformer: A only

(g) Ex. 4: MoEUT: Q+A

(h) Ex. 4: MoEUT: A only

Figure 43: Training Transformer and MoEUT models from scratch on DeepMind Math dataset arithmetic subset, with and without applying loss to the question part of the input. We show the residual erasure experiments here, with identical examples to Fig. 40. (a,e) We can see that if the model is trained with the question modeling enabled, it does not use its 2nd half of the layers, similarly to the LLMs. (b,f) If modeling the question only, the model uses significantly more layers. (c,d,g,h) MoEUT successfully uses more layers even when modeling the question, although modeling the answer only seem to help further (f). For more examples, please refer to Fig. 42.

