# OpenReview forum: "Do Language Models Use Their Depth Efficiently?"
_NeurIPS.cc/2025/Conference — NeurIPS 2025 poster_

### Official Review · Reviewer_TjFH · 2025-06-09

**Clarity:** 3
**Significance:** 3
**Originality:** 3
**Rating:** 4
**Confidence:** 3

**Summary:**

This paper investigates whether Transformers use depth efficiently, i.e., whether larger depth is utilized for higher-order computation or simply spreading computation. The authors analyze the residual stream in the LLaMA 3.1 model family. Through a series of empirical studies, they conclude that increased depth does not result in new computational capabilities but instead improves residual adjustments.

**Questions:**

1. Can the authors provide further justification or theoretical grounding for the layer contribution metrics used in the analysis?
2. Do the findings still hold for problems that cannot be solved by shallow models and inherently require deeper computation? For example, is a 10-layer model only able to solve problems whose solutions can be represented by models with effective depth no greater than, say, 5 layers? Or would it learn to utilize depth more effectively in this case?
3. Prior work [1] [2] suggests that CoT prompting improves reasoning by increasing the effective computational depth of LLMs. Do the conclusions of this paper contradict such findings? Or do they suggest that CoT compensates for poor depth utilization in standard forward passes?
4. Is the inefficiency in depth usage due to architectural limitations of Transformers (i.e., expressive capabilities) or to properties of the training process (e.g., optimization bias or data-induced inductive bias)?

[1] Feng et al. Towards revealing the mystery behind chain of thought: a theoretical perspective. NeurIPS 2023.


[2] Merrill et al. "The expressive power of transformers with chain of thought." ICLR 2024.

**Ethical Concerns:**

["NO or VERY MINOR ethics concerns only"]

**Final Justification:**

The paper reveals an interesting phenomenon regarding the depth usage of Transformers. The authors have addressed most of my concerns in the rebuttal. However, I remain unconvinced that the current evidence fully supports the conclusions, and I believe the paper would benefit from stronger empirical or theoretical support. Taking all these factors into account, I decide to set my score to 4.

**Limitations:**

See Weaknesses.

**Quality:**

2

**Strengths And Weaknesses:**

**Strengths**
1. The paper tackles a fundamental and timely research question: how effectively Transformers leverage their depth. This has direct implications for architecture design, interpretability, and scaling strategies.
2. The experiments are extensive. Multiple analysis techniques are employed to examine the effect of depth, and the results are presented clearly.

**Weaknesses**
1. The layer contribution metrics used in the analysis are introduced based on intuition, but their theoretical justification is limited. It would strengthen the paper to explain why these metrics are valid or whether they align with any existing formal frameworks.
2. The connection between depth utilization and task complexity is underexplored. The paper primarily evaluates tasks that could plausibly be solved by relatively shallow models. It remains unclear whether the conclusions would hold for tasks that inherently require deeper computation.

---

> ### Author Rebuttal · Authors · 2025-07-30
>
> We would like to thank the reviewer for the insightful review and for acknowledging the timeliness of our paper and the extensiveness of the experiments. Unfortunately, it is not possible to update the paper in the rebuttal period, so we are constrained to answer in text. Please see our responses below.
>
> > The layer contribution metrics used in the analysis are introduced based on intuition, but their theoretical justification is limited
> > Can the authors provide further justification or theoretical grounding for the layer contribution metrics used in the analysis?
>
> We are not sure what a satisfactory theoretical justification would look like in the eyes of the reviewer.
>
> Our motivation was the following: the layers in a pre-layernorm Transformer modify the residual by adding their output to it. If this output changes, it means that the layer’s computation changed. If this happens because of skipping a previous layer, it means that the output produced by the previous layer was used as an input for this layer, because the computation relied on it.
>
> > The connection between depth utilization and task complexity is underexplored. The paper primarily evaluates tasks that could plausibly be solved by relatively shallow models. It remains unclear whether the conclusions would hold for tasks that inherently require deeper computation.
>
> We chose the math domain because that is one of the tasks that we expect to require significant depth. Previous work, such as Sun et al. (2024): Transformer Layers as Painters, showed that GSM8K is the most sensitive to layer interventions among all tasks they tested. Could the reviewer provide a concrete task that is solvable with pretrained LLMs out of the box, and requires more depth than math? We are eager to explore such cases.
>
> > Do the findings still hold for problems that cannot be solved by shallow models and inherently require deeper computation? For example, is a 10-layer model only able to solve problems whose solutions can be represented by models with effective depth no greater than, say, 5 layers? Or would it learn to utilize depth more effectively in this case?
>
> This is an excellent question. We suspect that if we train the models on a synthetic task requiring 10 layers without modeling the input question, the model will utilize all 10 available layers. In fact, previous work, such as Csordás et al. (2021), "The Neural Data Router: Adaptive Control Flow in Transformers Improves Systematic Generalization," has already shown this. However, in Tab 4. Appendix, the authors also showed that it is possible to learn the same task with a much shallower model. This model relies on shortcuts, yet achieves 100% IID performance, albeit with low OOD generalization performance. In toy settings, it is possible to disentangle these two cases with carefully designed synthetic data, but it is not generally possible to do so for pre-trained LLMs, which are trained on most of the Internet. This was the primary motivation of our study: can we find evidence that LLMs break down problems into subproblems in a manner consistent with what we would expect from a generalizing solution? Unfortunately, we saw no evidence for this.
>
> Our hypothesis is that if we were to take the same task and combine it with language modeling, the model would likely use fewer layers for the task and dedicate the final layers to probability refinement. Similar effects in our preliminary study can be seen by comparing the Q and Q+A versions of the identical models in Fig 31 and Fig 32 in the Appendix: forcing the model to learn the question distribution seems to force the feature building stage to end earlier. Studying this systematically would be an interesting follow-up work.
>
> > Prior work [1] [2] suggests that CoT prompting improves reasoning by increasing the effective computational depth of LLMs. Do the conclusions of this paper contradict such findings? Or do they suggest that CoT compensates for poor depth utilization in standard forward passes?
>
> Our view is that CoT compensates for the poor depth utilization and lack of decomposition, which are among the main reasons for their success. However, CoT has its limitations. For example, there is no guarantee that the CoT trace contains the ideal decomposition, which guarantees that the model can use a fixed depth. Also, during pretraining, there is no CoT available, forcing the model to learn fixed-depth computation for all problems it sees. This can result in shortcut behaviors that must be unlearned during the post-training phase. It is unclear if this happens successfully. We believe an ideal model should be able to perform CoT, while also performing dynamic latent computation within its layers.
>
> > Is the inefficiency in depth usage due to architectural limitations of Transformers (i.e., expressive capabilities) or to properties of the training process (e.g., optimization bias or data-induced inductive bias)?
>
> This is an excellent question. In fact, we ask the same question while discussing the latent thinking methods in L306-308: “If the insensitivity of computation depth to input complexity is the consequence of the pretaining objective, such methods are fundamentally flawed. On the other hand, if the reason is the architecture, these approaches might provide the solution.“
>
> We hypothesize that it is at least partially caused by the next-token prediction loss, especially on the “impredictable” parts of the data that are not the result of a systematic computation, but should produce a complicated distribution of next tokens. Even if it is caused by the training process, it may be possible to construct architectures that behave more efficiently.
>
> We tried our best to resolve the concerns that the reviewer has raised. If the reviewer finds our response useful, please consider increasing the score. Thank you very much.

---

> > ### Comment · Reviewer_TjFH · 2025-08-06
> >
> > Thank you for the detailed rebuttal, which addresses part of my concerns.
> > Based on these clarifications, I will raise my score to 4.
> >
> > However, I still have some reservations about whether the current experimental results fully support the conclusions. A more fine-grained empirical evaluation would be helpful. For example, when evaluating deep and shallow models, it would be informative to separately consider tasks that are solved by neither model, only by the deep model, and by both models. Quantitative characterization in these categories, as I suggested in my comment about the underexplored connection between depth utilization and task complexity, could strengthen the conclusions. Alternatively, a theoretical analysis might also help justify the claims.

---

> > > ### Author Response · Authors · 2025-08-07
> > >
> > > We would like to thank the reviewer for increasing their score and acknowledging that our rebuttal addressed part of their concerns.

---

### Official Review · Reviewer_y38W · 2025-06-25

**Clarity:** 3
**Significance:** 3
**Originality:** 3
**Rating:** 4
**Confidence:** 4

**Summary:**

This paper investigates the depth efficiency of modern large language models through the lens of mechanistic interpretability. By inspecting the residual stream representations of LLMs through logit lens and by measuring their causal contributions to model outputs via causal intervention, they found that layers in the second half contribute much less than those in the first half. Moreover, for semantically more complex problems, LLMs fail to utilize more layers to solve them, suggesting a lack of compositionality in model reasoning. Another probing analysis suggests that deeper layers are not performing "new" operations compared to early layers. Together, these empirical results suggest that LLMs are not using deeper layers efficiently, which may also explains the diminishing return of scaling up transformer by adding more layers.

**Questions:**

See the weaknesses section.

**Ethical Concerns:**

["NO or VERY MINOR ethics concerns only"]

**Final Justification:**

The authors partially addressed my concerns during rebuttal, so I raised the significance score while keeping the others unchanged.

**Limitations:**

yes

**Quality:**

3

**Strengths And Weaknesses:**

Strengths:
* The relation between neural network model depth and performance is an important research problem, especially in the era of LLMs where scaling up model size often lead to improved general performance.

* The paper applied various mechanistic analysis tools to study LLM residual streams, and the experiments yield converging evidence that LLMs are not leveraging its deeper (especially second-half) layers efficiently.

* The paper presents an inspiring discussion section by drawing relations between their interpretability findings and LLM reasoning capabilities using Chain-of-Thought and Latent Thinking.

Weaknesses:
* While the analysis results presented in the paper are interesting, their empirical implications still seem vague to me. In particular, I found the argument that "the second half of the transformer models seem wasteful" a bit unsupported. According to Fig.4, without the second half layers, the model would yield an output distribution whose top-5 most likely token is very different from those predicted by the full model -- this should be considered as an essential step in computing the output probability distribution, as opposed to an incremental "refinement" step.

* I'm not fully understanding the logical relation between the claimed LLM depth inefficiency and the potential ineffectiveness of latent thinking methods. Even if non-recurrent transformers are inefficient in using deeper layers, a recurrent one pretrained from scratch may still evolve in a different way. The analyses in this paper cannot offer insights in answering this question.

---

> ### Author Rebuttal · Authors · 2025-07-30
>
> We would like to thank the reviewer for the insightful review and for finding our discussion section inspiring. Unfortunately, it is not possible to update the paper in the rebuttal period, so we are constrained to answer in text. Please see our responses below.
>
> > … I found the argument that "the second half of the transformer models seem wasteful" a bit unsupported. …
>
> We hypothesize that computing composite features for future predictions seems more useful than dedicating half of the layers to compute updates that are only useful for a single token. Additionally, inference typically uses nucleus sampling, discarding the tail of the probability distribution. Given that the tail is probably the most difficult to predict, it likely requires a significant amount of model capacity. It seems reasonable to assume that if this “distribution refinement” could be accomplished in fewer layers, and more layers could be devoted to computing useful features, the models could become more powerful predictors. Showing whether this computation is necessary or useful is an interesting future research direction.
>
> > I'm not fully understanding the logical relation between the claimed LLM depth inefficiency and the potential ineffectiveness of latent thinking methods. Even if non-recurrent transformers are inefficient in using deeper layers, a recurrent one pretrained from scratch may still evolve in a different way.
>
> We agree with the reviewer that our paper does not prove that latent thinking models are ineffective. In fact, in the paper we write: “If the insensitivity of computation depth to the input
> complexity is the consequence of the pertaining objective, such methods are fundamentally flawed. On the other hand, if the reason is the architecture, these approaches might provide the solution.” Unfortunately, we are unaware of large-scale pretrained recurrent depth models that we can test; thus, we did what we considered to be the next best experiment: a controlled study of MoEUT vs Transformers on toy tasks.
>
> We would also like to note that while not finding dynamic computation depth and decomposition in fixed-depth models does not invalidate recurrent thinking models, if we had found evidence that even fixed-depth non-recurrent models break down the problem dynamically in their layers, that would be a strong support for the latent reasoning models. Unfortunately, it seems that this is not the case.
>
> We tried our best to resolve the concerns that the reviewer has raised. If the reviewer finds our response useful, please consider increasing the score. Thank you very much.

---

### Official Review · Reviewer_WGWG · 2025-06-30

**Clarity:** 3
**Significance:** 3
**Originality:** 3
**Rating:** 5
**Confidence:** 4

**Summary:**

This paper investigates the relevance of different layers in a large language model when prompted to solve mathematical and more general reasoning tasks. Through a sequence of analyses involving the average norms and cosine similarities of terms which define the residual stream, causal interventions, linear mappings between residual streams of different models, the authors draw the main conclusion that the second half of the layers in a language model is used to make local changes to the residual stream required for accurately predicting the current token, with this half of layers having little influence on the prediction of future tokens. An analysis of the MoEUT architecture shows that universal transformers may not suffer from the same limitations.

**Questions:**

**Questions**

- While it was my impression that the observation of "the second half of the layers refining the current token's probability distribution" is framed somewhat negatively, could this not also be a positive property of non-universal transformers?
- One sentence in the abstract poses the question of whether transformers *create higher-order computations that are **impossible** in shallower models*. Do you have any formal results confirming that certain computations that you investigate are indeed **impossible** in shallower models?
- I must ask the authors what novel observations their analysis brings? I am aware that, even if not every observation is entirely novel, the analysis is valuable due to its scope and thoroughness, hence I am not interested in rejecting on the trivial ground of novelty. I am simply wondering which observation is entirely novel, given certain overlap of this work with at least that of *Lad et al., 2024*.


I do not doubt that we can lead a meaningful discussion after which I will consider increasing my score. I would in particular appreciate a discussion on the originality of your approach, seeing as it's one of the grading criteria.

**Ethical Concerns:**

["NO or VERY MINOR ethics concerns only"]

**Final Justification:**

I would in general appreciate more rigour in the paper, e.g. refraining from calling a computation "impossible" without a formal proof that it is in fact impossible. The same comment holds for the rebuttal. The authors mention that it is always possible to learn an N-way operation that solves any operation in few layers, but there is no mention of any formal guarantee confirming this statement.

That being said, I again state that I believe the analysis is interesting and could be impactful, and the authors could clarify certain important aspects of their work in the rebuttal. The observations, made on a range of LLMs and complex tasks. consistently point to the same conclusion, one which is currently not well-known in LLM literature.

**Limitations:**

yes

**Quality:**

3

**Strengths And Weaknesses:**

**Strengths**

- The paper presents a sequence of interesting observations which all point in coherence to the main conclusions, that (a) non-uniform transformers tend not to utilize the second half of the layers when predicting future tokens and that (b) the second half of the layers tends to be utilized to make smaller refinements of the residual stream necessary for predicting the current token.
- One must commend the sheer amount of experimental evidence the authors provide, all of which cohesively (although perhaps not unshakeably conclusively) points to the final drawn conclusions. This also causes me to believe that the paper is entirely technically solid. I find  zero reason to doubt any of the presented evidence.
- Observations made in this paper can an impact on future research on architectures, training objectives, and optimizers. I personally believe that the observations made in this work are valuable.
- The linear mapping of the residual streams from all of the layers of one model to all of the layers of another model is, i find, a very interesting experiment.

**Weaknesses**

- Some of the evidence presented is not entirely convincing. For example, while Figure 8 does show a diagonal pattern, I would be hesitant to draw too strong of a conclusion from it. I find that it is not entirely clear what it means to "stretch out" the same computation over several layers. Of course, I acknowledge the difficulty of providing exact explanations of the internal computations of large language models, and the sheer amount of partial evidence provided in this work does point to the conclusions drawn by the authors. I would just be cautious about stating that deeper models "stretch out" the same computation that shallower models perform over several layers, especially considering more formal results which provide results on the transformer depth necessary to perform certain computations (See e.g. the logarithmic dependence in Liu et al. 2023). Another claim that I find somewhat unconvincing is the statement that, paraphrasing lines 248-250, *The fact that MoEUT tends to make use of more depth in its computation than a non-universal transformer **confirms the advantage of shared-layer models.*** I do not believe that this fact really confirms anything.
- I believe that the text could be reformatted so that fewer references to sections and figures in the appendix are necessary. For example, in section 3.5., the four examples are not at all explained in the main text, being completely relegated to the appendix. I understand that, again, given the sheer amount of material the paper presents, it is difficult to adhere to the 9-page limit, but I still believe that there is room for improvement in this regard.
- I believe that a better overview and explanation of the MoEUT results would have been in order. It is rather unsurprising that a universal transformer utilizes its layers in a drastically different manner than a non-uniform transformer does. It would have been interesting if this higher layer utilization could be linked to higher extrapolation performance in a more systematic manner than just listing the one example of Mul/Div Multiple Longer.
- The claimed "refinement of the current token's probability distribution" sounds like a very interesting phenomenon, and I believe that it could have been addressed in more detail. But again I must acknowledge that one paper cannot do it all, so this weakness is rather to be taken as my opinion on what the ideal paper would do.
- As essentially every experimental analysis, this one could also benefit from a wider set of benchmarks. Additionally, I have to question its relevance, given that nowadays chain-of-thought based approaches seem to be the standard for the type of task the authors investigate.

---

> ### Author Rebuttal · Authors · 2025-07-30
>
> We would like to thank the reviewer for the insightful review and for acknowledging the diversity of our experiments that point to the same conclusion and their relevance for future research. Unfortunately, it is not possible to update the paper in the rebuttal period, so we are constrained to answer in text. Please see our responses below.
>
> > For example, while Figure 8 does show a diagonal pattern, I would be hesitant to draw too strong of a conclusion from it. I find that it is not entirely clear what it means to "stretch out" the same computation over several layers.
>
> We meant by “stretch out” that representations after the layers at the same relative position in the network map to each other the best. This means that similar computation is performed by both the deep and shallow models until the same percentile of the layers. An alternative would be to perform a chunk of computation corresponding to higher-level features that are impossible in shallow models. In such a case, these layers should exhibit high loss compared to any layer in the shallow model. We saw no evidence of such behavior.
>
> > I would just be cautious about stating that deeper models "stretch out" the same computation that shallower models perform over several layers, especially considering more formal results which provide results on the transformer depth necessary to perform certain computations (See e.g. the logarithmic dependence in Liu et al. 2023).
>
> We agree that certain computations require deeper models. However, they can often be “solved” on a limited range of data using shortcuts that do not generalize (see e.g. Table 4 in the appendix of Csordás et al. (2021): The Neural Data Router: Adaptive Control Flow in Transformers Improves Systematic Generalization). In LLMs, this is particularly challenging to detect due to the vast amount of training data. This was the primary motivation of our study: can we find evidence that the model adjusts its depth of computation based on problem complexity or operation ordering, or similar factors? Such a behavior would indicate decomposition and building on subresults. Unfortunately, we find no evidence for this.
>
> > The fact that MoEUT tends to make use of more depth in its computation than a non-universal transformer confirms the advantage of shared-layer models. I do not believe that this fact really confirms anything.
>
> We will change the wording of this sentence: MoEUT tends to utilize more depth in its computation than a non-universal transformer suggests that shared-layer models might have an advantage in this regard.
>
> The reason to believe that shared layers are better is that the knowledge can be easily transferred and reused in such models. Thus, it could be easier for the models to reuse already existing computations in the later layers, even if the gradient is dominated by the classification layer.
>
>
> > I believe that the text could be reformatted so that fewer references to sections and figures in the appendix are necessary.
>
> We will try our best to improve the readability of the final version of the paper.
>
> > I believe that a better overview and explanation of the MoEUT results would have been in order.
>
> We agree with the reviewer that more exploration of MoEUT would be helpful. However, for a fair comparison to MoEUTs, we would need a model trained on a similar scale and token budget as the LLMs we analyzed. Unfortunately, such a model doesn’t exist, and we do not have the budget to train it. Our MoEUT results aim to provide a preliminary study on a toy task, comparing it to a toy Transformer model trained on the same task.
>
> > The claimed "refinement of the current token's probability distribution" sounds like a very interesting phenomenon, and I believe that it could have been addressed in more detail.
>
> We agree with the reviewer that the details of this process are important to understand. However, we believe this is a research project in its own right, and we have left it for future work.
>
> > As essentially every experimental analysis, this one could also benefit from a wider set of benchmarks.
>
> We agree with the reviewer that more benchmarks are always beneficial, but we would also like to note that our paper offers a wide range of methods to test various aspects of the same phenomenon (future effect plots, integrated gradients, residual erasure). In terms of datasets, during the rebuttal period, we measured the future effects (Fig 3b.) on the Math dataset as well, and we obtained very similar results to the GSM8K in the paper. We will report this in the final version of our paper. Note that the toy datasets reported throughout various parts of the paper (arithmetic, multi-hop QA, DeepMind math) all exhibit a consistent behavior.
>
> > Additionally, I have to question its relevance, given that nowadays chain-of-thought based approaches seem to be the standard for the type of task the authors investigate.
>
> Chain-of-thought helps with the step-by-step computation and decomposition. We think this is the main reason behind its success. However, CoT has its limitations. For example, there is no guarantee that the CoT trace contains the ideal decomposition guaranteeing that the model can use fixed depth. Also, during pretraining, there is no CoT available, forcing the model to learn fixed-depth computation for all problems it sees. This can result in shortcut behaviors that must be unlearned during the post-training phase. It is unclear if this happens successfully. We believe an ideal model should be able to perform CoT, while also performing dynamic latent computation within its layers.
>
> > … "the second half of the layers refining the current token's probability distribution" is framed somewhat negatively, could this not also be a positive property of non-universal transformers?
>
> We hypothesize that computing composite features for future predictions seems more useful than dedicating half of the layers to compute updates that are only useful for a single token. Additionally, inference typically uses nucleus sampling, discarding the tail of the probability distribution. Given that the tail is probably the most difficult to predict, it likely requires a significant amount of model capacity. It seems reasonable to assume that if this “distribution refinement” could be accomplished in fewer layers, and more layers could be devoted to computing useful features, the models could become more powerful predictors. However, we agree with the reviewer that this is not a proven idea, and we acknowledge that the distribution refinement behavior might be beneficial in some way for the model. This is another interesting future research direction.
>
> > Do you have any formal results confirming that certain computations that you investigate are indeed impossible in shallower models?
>
> No, we do not have such results at present. However, it seems reasonable to assume that an N-step arithmetic operation requires at least N layers to learn a faithful elementwise computation that can generalize. Note that it is always possible to learn an N-way operation that solves any operation in a few layers, but such a solution is unlikely to generalize to slightly different scenarios. Finding evidence that distinguishes between these two scenarios is one of the motivations behind our study.
>
> Additionally, note that previous work, such as Sun et al. (2024), "Transformer Layers as Painters," demonstrated that, unlike many others, math-related tasks are sensitive to layer interventions across all datasets they tested. This is the reason why we based our investigation primarily on GSM8K.
>
> If the reviewer is aware of a task that has been formally proven to require deep models and is zero-shot doable by LLMs, we would be excited to test it.
>
> > I must ask the authors what novel observations their analysis brings? … I am simply wondering which observation is entirely novel, given certain overlap of this work with at least that of Lad et al., 2024.
>
> Our analysis has multiple novel factors. For example, we are not aware of any previous methods that are able to localize interactions between the layers directly, such as our interventions presented in Fig 3 and Fig 5. Also, we are not aware of any paper that studies whether the models vary the depth of their computation based on problem difficulty (Fig 7) or position in the computation graph (Fig 6, Fig 9, etc). Regarding the attention contribution, while the newest version of Lad et al, 2024 contains a more detailed analysis on attention (Fig. 6a), this was published in an update to the submission that happened after the NeurIPS deadline. Additionally, their attention analysis measures the attention value placed on the previous few tokens and does not directly assess the cross-position effect of different layers. We also show a connection between the diminishing importance of the later computation and the refinement behavior of the layers. Additionally, we are not aware of any previous work that tried to find a correspondence between layers in different depth networks.
>
> Overall, the goal of our paper is to provide a novel perspective, showing that LLMs do not appear to perform dynamic, problem-dependent decomposition in their latent computation. We are not aware of a study focusing on this problem.
>
> We tried our best to resolve the concerns that the reviewer has raised. If the reviewer finds our response useful, please consider increasing the score. Thank you very much.

---

> > ### Comment · Reviewer_WGWG · 2025-08-04
> >
> > Thank you very much for your detailed response. I believe now as I did when I wrote the review that this is an interesting study, and many of the analysis tools you presented here can be of value to LLM designers for improving the capabilities of their models.
> >
> > I would in general appreciate more rigour in the paper, e.g. refraining from calling a computation "impossible" without a formal proof that it is in fact impossible. The same comment holds for the rebuttal. You mention that it is always possible to learn an N-way operation that solves any operation in few layers, but there is no mention of any formal guarantee confirming this statement.
> >
> > That being said, I again state that I believe the analysis is interesting and could be impactful, and the authors could clarify certain important aspects of their work in the rebuttal. I will raise my score to a 5.

---

> > > ### Author Response · Authors · 2025-08-04
> > >
> > > We would like to thank the reviewer for acknowledging the usefulness of our method and for raising their score.
> > >
> > > We will change the wording of the paper to be less strong wherever no formal proof exists. Thank you for pointing this out.

---

### Official Review · Reviewer_Tdns · 2025-07-02

**Clarity:** 2
**Significance:** 2
**Originality:** 3
**Rating:** 4
**Confidence:** 4

**Summary:**

The paper asks whether modern pre-layernorm Transformer language models (Llama-3.1 and Qwen-3) truly exploit additional depth for new computations or simply spread the same work across more layers. Using a pack of residual-stream analyses and causal interventions—including layer-wise contribution analysis, causal layer-skipping, LogitLens & prediction-overlap, complexity probes with a depth-score metric, cross-model linear mapping, and an alternative-architecture (MoEUT) check—the authors uncover a sharp phase split: the first half of the network drives most residual change, while the second half mainly fine-tunes the current-token logits. These findings imply that stacked-Transformer LLMs under-utilize their later layers, devoting them to incremental logit sharpening rather than hierarchical composition, which helps explain diminishing returns from depth scaling and motivates architectures or objectives that make deeper networks more effective.

**Questions:**

Covered in the Strengths and Weaknesses section.

**Ethical Concerns:**

["NO or VERY MINOR ethics concerns only"]

**Final Justification:**

During the rebuttal the authors promised to add discussion of Peri-LN and other related work, and demonstrated that OLMo-2 (Post-LN-like) models show the same “late-layer under-use,” supporting the result’s generality. They also committed to fully labeled axes and legends, fixing readability concerns. Open questions, such as the precise cause of late-layer attenuation (residual growth vs. gradient decay) and the limited exploration of alternative architectures like MoEUT, remain for future work, yet they do not undermine the paper’s central empirical message.

**Limitations:**

yes

**Quality:**

2

**Strengths And Weaknesses:**

### Strengths
- Insightful multi-angle analysis: The paper fuses a diverse set of diagnostics. Since every tool points to the same phase-split conclusion, the evidence base is persuasive.
- Practical relevance for depth scaling: By demonstrating that later layers in very deep Transformers mostly perform incremental logit-sharpening, the study offers a explanation for the diminishing returns observed in depth-scaling curves. This insight can inform both model-design choices and future research on more depth-efficient objectives or architectures.

### Weaknesses
- Connection to the latest literature is thin: Several closely related lines of work are not discussed. Incorporating results from Mix-LN [1], Peri-LN [2], selective attention pruning [3], and Massive Activations [4] would situate the contribution more firmly within the current discourse and might even sharpen the interpretation of the phase-split finding.

- Architectural bias toward pre-LN models: All experiments use pre-layer-norm Transformers; it therefore remains unclear whether the conclusions extend to post-LN or other variants such as the OLMo architecture [5] or [2]. A small-scale replication on at least one non-pre-LN baseline, or a principled argument for transferability, would substantially strengthen the claim of generality.

- Interpretability of massive activations remains unexplored: Since residual-path signals can accumulate large magnitudes (as documented by Sun et al. [4]), tying the observed logit-sharpening behavior to activation growth patterns could provide a more mechanistic explanation of how and why later layers lose representational diversity.

- Clarity of key visualizations: Several figures—most notably the depth-score heat-map—lack fully labeled axes, color bars, or legends, forcing readers to consult the appendix for basic interpretive cues. Self-contained captions would improve accessibility.

- Limited evidence for alternative architectures: The MoEUT comparison relies on only one or two checkpoints. Extending the study to a broader range of shared-parameter or other architectures would give greater empirical weight to the “look beyond pure stacking” recommendation.

- Root cause of logit-smoothing remains open: The paper compellingly documents what happens in the back half of the network but stops short of explaining why. Some competing hypotheses are neither disentangled nor tested experimentally such as  (1) the next-token objective provides diminishing gradient signal at depth, and (2) architectural bottlenecks curb the formation of new representations

[1] Li, Pengxiang, Lu Yin, and Shiwei Liu. "Mix-ln: Unleashing the power of deeper layers by combining pre-ln and post-ln." ICLR2025.
[2] Kim, Jeonghoon, et al. "Peri-LN: Revisiting Normalization Layer in the Transformer Architecture." Forty-second International Conference on Machine Learning. 2025.
[3] He, Shwai, et al. "What matters in transformers? not all attention is needed." arXiv preprint arXiv:2406.15786 (2024).
[4] Sun, Mingjie, et al. "Massive activations in large language models." COLM2024.
[5] OLMo, Team, et al. "2 OLMo 2 Furious." arXiv preprint arXiv:2501.00656 (2024).

---

> ### Author Rebuttal · Authors · 2025-07-30
>
> We would like to thank the reviewer for the insightful review and appreciating the papers insightful analysis and practical relevance. Unfortunately, it is not possible to update the paper in the rebuttal period, so we are constrained to answer in text. Please see our responses below.
>
> > Incorporating results from ... would situate the contribution more firmly within the current discourse and might even sharpen the interpretation of the phase-split finding.
>
> We will add the discussion of these papers to the related work section and relate them to our findings. For example, Peri-LN [2] shows that their method has increased angular distance between the representations of different layers, indicating that it might use deeper layers more efficiently. However, if the reviewer is suggesting measuring the effects that such changes have on the model's behavior and our analysis, we would like to note that our study primarily focuses on pre-trained models, and we believe that a reliable signal about multi-step processing and reasoning can only be obtained from well-trained models. Thus, such a study needs to be conducted by one of the big industry labs, which can afford to systematically train a series of models with these modifications.
>
> > Architectural bias toward pre-LN models: All experiments use pre-layer-norm Transformers; it therefore remains unclear whether the conclusions extend to post-LN or other variants such as the OLMo architecture [5] or [2]. A small-scale replication on at least one non-pre-LN baseline, or a principled argument for transferability, would substantially strengthen the claim of generality.
>
> We mainly analyze pre-LN models (x + f(LN(x))), because the majority of freely available models are pre-LN. We agree that the PostLN from the original Vaswani et al. (2017) paper, in the form of LN(x+f(x)) might have an impact on our findings and improve the feature building stages of the model. However, we are not aware of any successful LLMs using the PostLN layout. OLMo is x + LN(f(x)), which looks closer to PreLN than to PostLN; thus, the similarity in behavior is expected.
>
> In the final version of the paper, we will add an analysis of the OLMo 2 models. We ran these experiments during the rebuttal. The differences from the current findings are minimal. OLMo 2 7B shows an interesting pattern, where a small "triangle" appears in the similarity maps, indicating that certain layers are composed with each other but collectively have no effect on the later computations. The lack of influence of the late layers remains evident in this model. OLMo2 13B is basically indistinguishable from Qwen 3 8B (Fig. 18. a). OLMo 2 32B is similar to Qwen 3 32B, with slightly better layer usage.
>
> Unfortunately, we do not see a way to report our plots in the rebuttal without violating the rebuttal rules. If the reviewer has a suggestion, please let us know.
>
> > Interpretability of massive activations remains unexplored: Since residual-path signals can accumulate large magnitudes (as documented by Sun et al. [4]), tying the observed logit-sharpening behavior to activation growth patterns could provide a more mechanistic explanation of how and why later layers lose representational diversity.
>
> We agree with the reviewer that one possibility for why the later layers have less effect on future computations might be that the residuals are growing. But there are alternative explanations. For example, since the gradients of the later layers might be directly dominated by the classifier layer’s gradient distributed through he residual, instead of the higher-order effects through composing layers. Exploring the detailed mechanism behind these effects is an important future direction and is probably worth its own study.
>
> > Clarity of key visualizations: Several figures—most notably the depth-score heat-map—lack fully labeled axes, color bars, or legends, forcing readers to consult the appendix for basic interpretive cues. Self-contained captions would improve accessibility.
>
> We will improve this for the final version of the paper.
>
> > Limited evidence for alternative architectures: The MoEUT comparison relies on only one or two checkpoints. Extending the study to a broader range of shared-parameter or other architectures would give greater empirical weight to the “look beyond pure stacking” recommendation.
>
> MoEUT served only as an outlook for possible future architectures. A more significant issue with the MoEUT experiments is that we lack the compute resources to fully train a base MoEUT model on a scale comparable to the Llama and Qwen models; thus, we cannot be certain of the reasons for the differences. We are not aware of any shared-layer pretrained model, but in the case such a model exists, we would be very interested in analyzing it.
>
> > Root cause of logit-smoothing remains open: The paper compellingly documents what happens in the back half of the network but stops short of explaining why. Some competing hypotheses are neither disentangled nor tested experimentally such as (1) the next-token objective provides diminishing gradient signal at depth, and (2) architectural bottlenecks curb the formation of new representations
>
> Our paper aims to highlight that current LLMs often employ a fixed number of layers, regardless of problem complexity, and a significant portion of these layers fail to build reusable features. We analyze this effect in multiple ways throughout the paper (future effect plots, integrated gradients, residual erasure, depth score, etc). This analysis already barely fits in a single paper. While we agree with the reviewer that it is an important direction to determine the mechanistic reason behind the observed behaviors, we think that expecting to solve such a complex problem in a single paper is unreasonable.
>
> We tried our best to resolve the concerns that the reviewer has raised. If the reviewer finds our response useful, please consider increasing the score. Thank you very much.

---

> > ### Comment · Reviewer_Tdns · 2025-08-06
> >
> > I appreciate the authors’ detailed responses. Although some of the answers do not fully address the questions, I believe the in-depth discussion will be valuable for future research in this field. Therefore, I am raising my original rating.

---

> > > ### Author Response · Authors · 2025-08-07
> > >
> > > We would like to thank the reviewer for increasing their score and for their insightful review.

---

### Official Review · Reviewer_X1zR · 2025-07-03

**Clarity:** 2
**Significance:** 3
**Originality:** 2
**Rating:** 4
**Confidence:** 3

**Summary:**

This paper investigates whether deeper layers in modern LLMs contribute to more complex computations or simply distribute the same processes across more layers. Overall, their results suggest that current deep LLMs do not use additional layers to perform fundamentally new computations. Instead, depth is primarily used for finer adjustments of representations, possibly explaining diminishing performance returns from simply increasing model depth. The authors analyze the residual stream and layer contributions, finding that layers in the second half of the network contribute much less than the first half. Additionally, skipping later layers has minimal effect on future predictions while they don’t find evidence that deeper models use additional depth to compose multi-hop reasoning or complex intermediate steps. Their findings also suggest that deeper models spread existing computations over more layers instead of introducing new types of computation.

**Questions:**

- The behavior of the Qwen models appears to differ somewhat from that of the Llama models (e.g., Figure 15, Figure 18e). Could the authors offer any explanations for these differences? It might also be valuable to analyze additional model families to better understand whether the observed patterns are consistent across architectures.

- Are there notable differences in layer-wise behavior across different prompts, datasets, or tasks? While the paper includes some experiments on datasets like MQuAKE and MATH, several analyses rely on only 10 sampled prompts from GSM8K. It could strengthen the findings to include a broader range of prompts and datasets to assess the robustness and potential variability of the results.

- In the discussion in Section 5, the authors suggest that the second half of the layers might be wasteful, as these layers primarily perform distribution matching. However, could this distribution matching still be necessary, given that the prediction distribution halfway through the network remains substantially different from that of the final layer?

**Ethical Concerns:**

["NO or VERY MINOR ethics concerns only"]

**Final Justification:**

The authors have addressed some of my concerns during the rebuttal, particularly by running experiments on additional data and models. However, certain claims still seem somewhat strong, as some models exhibit differing behavior without clear supporting explanations (e.g., Qwen3-32B). Nonetheless, the overall analysis remains interesting and valuable, and I continue to recommend acceptance.

**Limitations:**

yes

**Quality:**

3

**Strengths And Weaknesses:**

Strengths
- The paper provides a thorough analysis of depth-wise computations in large language models, approaching the problem from several perspectives
- The insights provided in the paper have practical implications for future research and model design.

Weaknesses:
- The experimental evaluation relies on relatively small sample sizes and limited datasets (e.g. on some experiments, 10 random examples from GSM8K), which may constrain the generalizability of the findings.
- Similarly, the analysis is limited to only the Llama and Qwen model families, leaving open the question of whether the results generalize to other architectures.

---

> ### Author Rebuttal · Authors · 2025-07-30
>
> We would like to thank you for your insightful review and appreciate that you felt that the work had thorough analysis and practical implications. Unfortunately, it is not possible to update the paper in the rebuttal period, so we are constrained to answer in text. Please see our responses below.
>
> > ... evaluation relies on relatively small sample sizes and limited datasets ...
>
> > ... It could strengthen the findings to include a broader range of prompts and datasets
>
> While it is true that the main experiments are conducted on GSM8K, our other experiments involve simple arithmetics, multi-hop questions, and DeepMind Math, yielding consistent findings. We chose GSM8K based on the findings of Sun et al. (2024), "Transformer Layers as Painters," which showed that GSM8K is the most sensitive to layer interventions among all tasks they tested.
>
> In the final version of the paper, we will include experiments in the appendix showing an equivalent of Fig 2., Fig. 3 and Fig. 4 for:
> 1. 50 examples: This will show that 10 examples is sufficient to see the general trend.
> 2. 10 examples from the MATH dataset: This will show robustness to the dataset in the math domain.
>
> We have already conducted these future effect experiments during the rebuttal period, and we can confirm that they are consistent with the paper's findings. Unfortunately, due to the new NeurIPS rules, we don't see a way to upload these results.
>
> > ... the analysis is limited to only the Llama and Qwen model families ...
>
> We agree with the reviewer that having more results is always better. We would like to note that we already struggle with presenting all the model variants we tested in a comprehensive manner. We reported 3 sizes of both model families. Additionally, we also report results on the instruction-tuned Llama 70B (Fig. 19).
>
> Nevertheless, in the final version of the paper, we will add an analysis of OLMo 2. The differences from the current findings are minimal. OLMo 2 7B shows an interesting pattern, where a small "triangle" appears in the similarity maps, indicating that certain layers are composed with each other but collectively have no effect on the later computations. The lack of influence of the late layers remains evident in this model. OLMo2 13B is basically indistinguishable from Qwen 3 8B (Fig. 18. a). OLMo 2 32B is similar to Qwen 3 32B, with slightly better layer usage.
>
> > The behavior of the Qwen models appears to differ somewhat from that of the Llama models (e.g., Figure 15, Figure 18e). Could the authors offer any explanations for these differences?...
>
> We believe that the model's behavior is not only dependent on the model architecture, but also on the details of the training setup, including initialization and optimization setup. These training details are usually not public. Understanding what causes such behavior requires an extensive study, and we expect that fully understanding the effects could take multiple papers.
>
> Out of the models we tested, Qwen 32B seems to be somewhat different from the others: instead of not using the late layers for features useful for future predictions, it doesn't use the early layers. This also shows that it is possible to have a well-performing model without most of the late layers being dominated by the loss-refining behavior. Based on Fig. 22i, we can see that the logit refinement behavior is more compressed, but consistent with the previous findings: the predictions start to improve when feature building stops (~layers >50).
>
> > Are there notable differences in layer-wise behavior across different prompts
>
> Yes. The individual examples (prompts) exhibit different behaviors, but the general trends remain consistent. For example, Fig. 6 shows some of the variations in the processing behavior between two prompts. This is why we take the maximum over multiple different prompts. In the final version of the paper, we will include a few individual examples in the appendix as illustrations.
>
>
> > In the discussion in Section 5, the authors suggest that the second half of the layers might be wasteful, as these layers primarily perform distribution matching. However, could this distribution matching still be necessary, given that the prediction distribution halfway through the network remains substantially different from that of the final layer?
>
> We agree with the reviewer that such refinement might be necessary. However, we are unsure about the number of layers necessary for this. For example, Figs. 18e and 22i,j show that Qwen 3 32B is able to perform refinement in a very few layers compared to other models (this model suffers from a different issue: not utilizing its early layers for feature building).
>
> In general, this refinement behavior may be a consequence of the next-word prediction objective. However, we believe that architectural changes might influence the extent to which it dominates the network. It seems reasonable to assume that if this “distribution refinement” could be accomplished in fewer layers, and more layers could be devoted to computing useful features, the models could become more powerful predictors. Showing whether this computation is necessary or useful is an interesting future research direction.
>
>
> We tried our best to resolve the concerns that the reviewer has raised. If the reviewer finds our response useful, please consider increasing the score. Thank you very much.

---

> > ### Comment · Reviewer_X1zR · 2025-08-04
> >
> > I appreciate the clarifications provided by the authors, which addressed several of the questions and concerns I initially raised. I will tentatively maintain my positive score.
> >
> > Regarding the results on Qwen3 32B, it was noted that the model appears to underutilize its earlier layers for feature building. Based on Figure 18e, is it correct to interpret that most of the feature construction occurs around layers 35–45? If so, do the authors have any insights into the role or nature of the computations happening in the earlier layers in this case?

---

> > > ### Author Response · Authors · 2025-08-04
> > >
> > > We would like to thank the reviewer for acknowledging that our rebuttal addressed several of their concerns.
> > >
> > > We think the correct interpretation of Fig. 18e is that layers <35 are still useful (they have high influence on late layers), but their influence seems to be largely independent of each other until layer 35. From layer 35, the network starts using these mostly independent outputs of the early layers to build higher-order features. This means that the first step of computation of the feature might be in an early layer, but the higher-order compositions happen in later layers. Interestingly, there is only a tight band of layers that are both strongly influenced by early layers, and also strongly influence late layers: layers 35-50 (visible in the figure as a vertical bar that extends to the right side and also not part of the bottom right triangle tip). Layers >50 have low influence on the future layers (the bottom right triangle tip), except for the last 1-2, similarly to the late layers of other models.
> > >
> > > We would also like to note that this early independent feature building stage seems to be present in all bigger models, albeit to a much smaller extent: LLama 3.1 70B and 405B also have a similar, but smaller triangle on the top-left, as well as Qwen 3 14B. It is minimally visible in Qwen 3 8B as well.
> > >
> > > This behavior might suggest that the first few layers are bottlenecked by attention, and a substantial amount of information gathering is necessary before the feature building can begin. The surprising part for us is the extent to which this dominates Qwen 3 32B.
> > >
> > > One of the goals of our study is to raise awareness of these effects in Transformers, and inspire future studies on how to structure models more effectively, hopefully building higher-level features that are more context-dependent, but also for accelerating models. Here are a couple of examples: the independence of the late layers suggests that it might be possible to replace them with fewer, wider layers that are more parallelizable. The early layers might require "more" attention and "less" MLPs. The width of the residual might bottleneck this early, information gathering stage. Our methods also provide a diagnostic tool that can be used to measure layer interactions directly.

---

### Decision · Program_Chairs · 2025-09-17

**Decision:**

Accept (poster)

**Comment:**

This paper presents a comprehensive and insightful investigation into how large language models utilize their depth, a timely and significant question for the field. The reviewers reached a strong consensus on the paper's primary strengths, consistently praising the thorough, multi-faceted analysis that cohesively points to the conclusion that deeper models, particularly in their later layers, are underutilized for complex, compositional computations and instead focus on refining the current token's probability distribution. This central finding, supported by a diverse set of experiments including causal interventions and cross-model analysis, was potentially considered to have valuable practical implications for future research.

The review process also highlighted several key limitations. A recurring concern was the generalizability of the findings beyond the Llama and Qwen families and on a wider range of tasks and larger data samples. Reviewers also noted that while the paper compellingly documents the phenomenon of late-layer underutilization, it stops short of providing a deep mechanistic explanation for its root cause. The authors have constructively engaged with this feedback and, in their rebuttal, committed to addressing several of these points in the final version by including experiments on the OLMo model family, expanding their analysis to the MATH dataset, and improving the clarity of key visualizations. While open questions remain for future work, the paper's core empirical contributions are solid and offer a novel, valuable perspective on the functional dynamics of model depth.